# Rab5 and Alsin regulate stress-activated cytoprotective signaling on mitochondria

FoSheng Hsu[1], Stephanie Spannl[1], Charles Ferguson[2], Anthony A Hyman[1], Robert G Parton[2,3], Marino Zerial[1]*

[1]Max Planck Institute of Molecular Cell Biology and Genetics, Dresden, Germany; [2]Institute for Molecular Bioscience, University of Queensland, Brisbane, Australia; [3]Centre for Microscopy and Microanalysis, University of Queensland, Brisbane, Australia

**Abstract** Mitochondrial stress response is essential for cell survival, and damaged mitochondria are a hallmark of neurodegenerative diseases. Thus, it is fundamental to understand how mitochondria relay information within the cell. Here, by investigating mitochondrial-endosomal contact sites we made the surprising observation that the small GTPase Rab5 translocates from early endosomes to mitochondria upon oxidative stress. This process is reversible and accompanied by an increase in Rab5-positive endosomes in contact with mitochondria. Interestingly, activation of Rab5 on mitochondria depends on the Rab5-GEF ALS2/Alsin, encoded by a gene mutated in amyotrophic lateral sclerosis (ALS). Alsin-deficient human-induced pluripotent stem cell-derived spinal motor neurons are defective in relocating Rab5 to mitochondria and display increased susceptibility to oxidative stress. These findings define a novel pathway whereby Alsin catalyzes the assembly of the Rab5 endocytic machinery on mitochondria. Defects in stress-sensing by endosomes could be crucial for mitochondrial quality control during the onset of ALS.
DOI: https://doi.org/10.7554/eLife.32282.001

## Introduction

Mitochondria, the organelle for cellular metabolism and ATP production, play an essential role in a number of other cellular processes such as calcium signaling, lipid synthesis and trafficking, metabolite transport, apoptosis, and reactive oxygen species (ROS) production in the cell (*Mesmin, 2016*; *Ott et al., 2007*; *Rizzuto et al., 2012*). Many of these processes necessitate communication with other cellular compartments. For example, membrane contact sites (MCS) between endoplasmic reticulum (ER) and mitochondria are important for $Ca^{2+}$ and lipid transfer (*de Brito and Scorrano, 2008*), mitochondria fission (*Friedman et al., 2011*), and regulation of apoptosis (*Prudent et al., 2015*). Lipid droplets and peroxisomes interact with mitochondria to regulate fatty acid oxidation (*Cohen et al., 2014*; *Pu et al., 2011*). Oxidized and damaged proteins can be selectively delivered to peroxisomes and lysosomes via mitochondrial-derived vesicles (*Sugiura et al., 2014*). These examples demonstrate an extensive functional interplay between organelles, either directly via MCS and/or indirectly via vesicular intermediates. However, the underlying molecular mechanisms remain poorly understood and, in particular, the functional relationship between mitochondria and the endocytic system is largely unexplored.

The endocytic pathway is responsible for maintaining cellular homeostasis by internalizing, sorting, recycling and/or degrading distinct types of cargo molecules (*Huotari and Helenius, 2011*). Rab GTPases serve as molecular signatures on endosomes, regulating their biogenesis and functions (*Pfeffer, 2017*; *Zerial and McBride, 2001*; *Zhen and Stenmark, 2015*). Ligand-receptor complexes at the plasma membrane (PM) are internalized into early endosomes (EE) marked by small GTPase Rab5, followed by either recycling to the PM via Rab4 and Rab11-positive recycling endosomes (RE),

*For correspondence:
zerial@mpi-cbg.de

**eLife digest** The inside of a human cell is divided into compartments called organelles, which are surrounded by membranes. Each organelle plays a specific role in keeping the cell healthy and also has unique mix of molecular markers on its surface. These markers allow other molecules to identify the different organelles, meaning that specific organelles can communicate with each other and coordinate their activities. One way that organelles can do this is via so-called membrane contact sites, which are small areas where the compartments' outer membranes come close together.

Mitochondria are organelles that release energy inside human cells. These compartments also work to keep the levels of toxic chemicals called reactive oxygen species in the cell within a safe range. This is important because cells can die if these levels become too high – a state known as oxidative stress. Mitochondria also communicate with other organelles called endosomes, which receive materials from the cell surface, sort and direct them to different destinations throughout the cell.

In many diseases affecting the nervous system, the mitochondria and endosomes in nerve cells do not work properly. These cells also have higher than normal levels of oxidative stress. Hsu et al. therefore wanted to find out if mitochondria and endosomes worked together to help cells to cope with this kind of stress.

Hsu et al. triggered oxidative stress in human cancer cells by exposing them first to a dye that stained the mitochondria and then to intense light. In stressed cells, a subset of endosomes called early endosomes formed many more membrane contact sites with mitochondria than in non-stressed cells. At the same time, the protein Rab5, usually found on early endosomes, relocated to the surface of mitochondria. Human cells previously engineered to produce larger than normal amounts of Rab5 were also more likely to survive oxidative stress. These experiments suggested that early endosomes cooperate with mitochondria, via Rab5, to protect cells from oxidative stress.

So, how is Rab5 relocated to mitochondria? Hsu et al. searched for activators of Rab5 and found that Alsin also migrated to mitochondria in stressed cells. The gene for Alsin is also mutated in amyotrophic lateral sclerosis (ALS), a degenerative nerve disorder that remains poorly understood. Next, Hsu et al. deleted the gene for Alsin from human stem cells growing in the laboratory and coaxed these cells into becoming nerve cells. Experiments with these cells showed that the absence of Alsin prevented Rab5 from moving to the mitochondria. Nerve cells lacking Alsin were also more susceptible to oxidative stress than normal cells.

Together, these findings show that early endosomes work with mitochondria to sense and ward off oxidative stress. They also reveal an unexpected connection between this process and a gene mutated in ALS. Further experiments are now needed to explore if problems with endosomes or mitochondria, and specifically with molecules like Alsin and Rab5, are responsible for other neurodegenerative disorders, like Parkinson's disease and Huntington's disease.
DOI: https://doi.org/10.7554/eLife.32282.002

or routed to late endosomes (LE) and lysosomes for degradation. The latter process occurs via the conversion of Rab5-positive EE into Rab7-positive LE (*de Renzis et al., 2002*; *Rink et al., 2005*; *Sönnichsen et al., 2000*; *Ullrich et al., 1996*). Rab proteins can thus dynamically associate with the membranes, conferring functional plasticity to organelles. On these endosomal membranes, Rab proteins recruit a plethora of effectors for membrane tethering and fusion, cargo sorting and signaling (*Sorkin and von Zastrow, 2009*; *Stenmark, 2009*; *Zerial and McBride, 2001*). For example, EEA1 is a dimeric coiled-coiled Rab5 effector protein that tethers two vesicles to allow efficient fusion between Rab5-harbouring membranes (*Murray et al., 2016*; *Ohya et al., 2009*). Other Rab5 effectors such as APPL1 are involved in regulating metabolic and inflammatory responses (*Schenck et al., 2008*; *Wen et al., 2010*). Rab activation, and thus stabilization after recruitment, on the membrane requires guanine nucleotide exchange factors (GEFs) (*Blümer et al., 2013*). Rab5 GEFs constitute a family of VPS9 domain-containing proteins, including Rabex-5 (*Horiuchi et al., 1997*), RME-6 (*Sato et al., 2005*), amyotrophic lateral sclerosis protein 2 (ALS2/Alsin) (*Otomo et al., 2003*), and mammalian Ras and Rab interactor 1, 2, 3 (Rin1-3) (*Hu et al., 2005*). The rationale behind

this complexity is that Rab5 must be specifically regulated by different GEFs in space and time. In this respect, the function of many Rab5 GEFs remains unclear.

Physical interactions between the endosomal machinery and mitochondria serve important functions in cell homeostasis, repair and apoptosis. For example, transfer of iron (*Das et al., 2016*; *Sheftel et al., 2007*) and cholesterol (*Charman et al., 2010*) from endosomes to mitochondria is mediated by physical interactions between the two organelles. Another classical example of mitochondria-endolysosome interactions is autophagy. Autophagy is a clearance mechanism whereby cells identify defective organelles following damage or stress, and eliminate them via the formation of an autophagosome and subsequent fusion with lysosomes (*Mizushima and Levine, 2010*). The mechanism of degrading mitochondria has been termed macroautophagy or mitophagy. Intriguingly, the expression of pro-apoptotic factors such as canonical BH3-only proteins drive Rab5- and Rab7-positive endolysosomes into inner mitochondrial compartments via a pathway that appears to differ from autophagy/mitophagy (*Hamacher-Brady et al., 2014*). Interestingly, our previous conducted genome-wide RNAi screen of endocytosis (*Collinet et al., 2010*) revealed that ~8% of the hit genes had mitochondrial-related functions, pointing at hitherto unexplored molecular connections between the endosomal system and mitochondria. This led us to hypothesize that other mitochondrial functions may be regulated by endocytic components.

Here by exploring the interactions between EE and mitochondria, we unexpectedly found that upon laser- or chemically induced oxidative stress in mammalian cells, mitochondria outer membrane permeabilization (MOMP) releases cytochrome c, and concomitantly, triggers the assembly of the Rab5 machinery on mitochondria, in a process which is reversible and independent of mitophagy. Remarkably, we found that the Rab5 GEF responsible for Rab5 activation is ALS2/Alsin, which is necessary for efficient Rab5 recruitment to mitochondria. Our findings suggest that the Rab endocytic machineries interact with mitochondria during oxidative stress as a cytoprotective mechanism, with important implications for amyotrophic lateral sclerosis (ALS) and other neurodegenerative diseases.

## Results

### Inter-organelle contacts between endosomes and mitochondria

We first explored the potential physical link between the endosomal system and mitochondria at steady state. HeLa cells stably expressing TagRFP-MTS (mitochondria targeting sequence) (*Takeuchi et al., 2013*) were incubated with two types of endocytic cargo, Alexa-488-conjugated transferrin (Tfn) or epidermal growth factor (EGF) for 10 min at 37°C, to visualize the endocytic/recycling and degradative pathway, respectively. Cell were then fixed and imaged via confocal microscopy. We observed a subset of endosomes that appeared to partially overlap with, or were in close proximity to, mitochondria (*Figure 1A,B*). To avoid potential morphological changes induced by the TagRFP-MTS overexpression, we also performed quantitative measurements in cells stained for mitochondria with MitoTracker-Red CMXRos (Mito-Red) and labeled with Tfn-488 and EGF-488 for 10 and 60 min at 37°C. All acquired images were subjected to chromatic shift correction, deconvolution, and localization analysis (MotionTracking) based on subtraction of random colocalization (*Kalaidzidis and Zerial, 2015*). Tfn and EGF were efficiently internalized at 10 min, and further sorted to distinct perinuclear compartments representing RE (*Maxfield and McGraw, 2004*) and LE/lysosomes, respectively, at 60 min (*Figure 1C*). In both time points, Tfn-containing endosomes consistently exhibited higher colocalization with Mito-Red (*Figure 1D*) compared to EGF-structures, despite both having similar signal intensities (*Figure 1E*), suggesting that the interactions with mitochondria may preferentially involve early/recycling endosomal structures.

Given that early endosomes are motile and omnipresent in the cytoplasm, we next asked whether the observed physical proximity with mitochondria may reflect real MCS or be simply due to random chance (*Kalaidzidis and Zerial, 2015*). To further assess the interactions, we monitored the dynamics of organelle contacts during early endocytic events by live-cell imaging. Cells were incubated with Tfn-488 for 1 min to label EE and immediately imaged using a spinning disk confocal microscope. A number of Tfn-containing endosomes and mitochondria labeled by TagRFP-MTS were observed in close proximity, suggesting possible interactions (*Video 1*). Some endosomal vesicles remained in close contact with mitochondria for 3–5 min and, interestingly, we could also observe interactions that were followed by fission-like events (*Video 2*). Our data suggest that the occurrence of physical

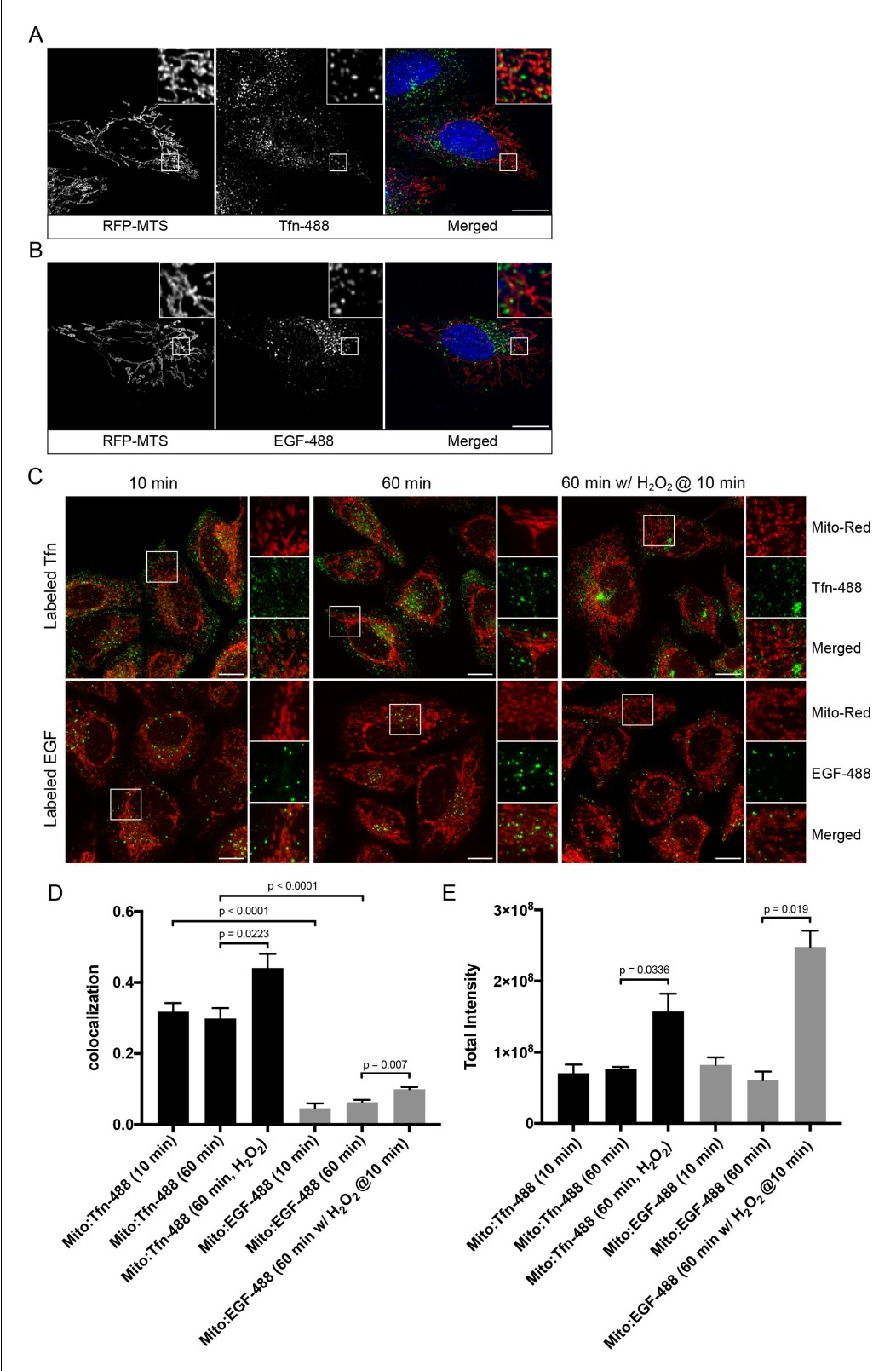

**Figure 1.** Endosomal contacts with mitochondria. (**A**) and (**B**) HeLa cells were transfected with TagRFP-MTS (mitochondria targeting sequence) and labeled with either Alexa-488 transferrin (Tfn) or epidermal growth factor (EGF) at 37°C for 10 min, respectively. (**C**) HeLa cells stained with Mito-Red were labeled with Tfn-488 or EGF-488 at 37°C for 10, 60 min, or treated with 250 µM $H_2O_2$ after 10 min labeling followed by additional 50-min incubation. Inset regions are shown. Scale bars, 10 µm. (**D**) Colocalization analysis based on subtraction of random colocalization was performed

*Figure 1 continued on next page*

*Figure 1 continued*

between Mito-Red and Tfn-488 or EGF-488 from images obtained in *Figure 1C*, *n* = 50. (E) Quantification of total intensity per cell, *n* = 50. Error bars represent SEMs. p Values based on two-tailed t-tests.

DOI: https://doi.org/10.7554/eLife.32282.003

The following source data is available for figure 1:

**Source data 1.** Numerical data corresponding to the graphs presented in *Figure 1D,E*.

DOI: https://doi.org/10.7554/eLife.32282.004

contacts between EE and mitochondria observed in fixed and live cells may reflect *bona fide*, albeit transient interactions, as suggested previously (*Das et al., 2016*; *Sheftel et al., 2007*).

## Acute mitochondrial stress recruits Rab5 and Rab5-positive endosomes to mitochondria

Given the key role of mitochondria in sensing and responding to oxidative stress, we asked whether acute perturbation on mitochondria may affect endosomes-mitochondria interactions. For this, we used HeLa cells stably expressing GFP-Rab5 under its endogenous promoter with a bacterial artificial chromosome (BAC) transgene (BAC GFP-Rab5) (*Villaseñor et al., 2015*). These cells were validated based on the lack of detectable alterations on endocytic trafficking. Live-cell imaging of BAC GFP-Rab5 expressing RFP-MTS also confirmed the occurrence of Rab5-positive EE (>200 nm) in close contact with mitochondria (*Video 3*).

To validate the results with the ectopically expressed mitochondrial marker, we tested mitochondrial-selective dyes in our live-cell imaging. Unexpectedly, we found that photoirradiation in cells labeled with Mito-Red not only robustly induced alterations in mitochondrial morphology but also the appearance of GFP-Rab5 signals around mitochondria (*Figure 2A*). This is consistent with the fact that certain rosamines and rhodamine-derived dyes used to assay mitochondrial functions possess photosensitizing properties (*Hsieh et al., 2015*). Specifically, Mito-Red has been used to perturb mitochondrial function (*Minamikawa et al., 1999*). Therefore, we hypothesized that the change in Rab5 localization may be a consequence of alterations in mitochondrial function. Consistent with previous reports, low exposure of Mito-Red labeled cells with a 561 nm laser (~5 J/cm$^2$) caused a decrease in mitochondrial and an increase in cytoplasmic signal, indicative of MOMP, and accompanied by a globular swelling of mitochondria within min (*Figure 2A*, *Video 4*). Remarkably, laser treatment on Mito-Red-labeled cells resulted in the translocation of GFP-Rab5 to mitochondria, marked by increased colocalization with Mito-Red compared to the untreated (*Figure 2B*). Furthermore, the frequency of GFP-Rab5 EE in close proximity to the stressed mitochondria increased (*Figure 2A*, Post-laser arrowheads). This effect was specific to Mito-Red treatment, because cells labeled with MitoTracker Green FM or transfected with RFP-MTS retained their tubular mitochondrial structures under the same laser treatment and did not recruit GFP-Rab5 to the mitochondria (*Figure 2—figure supplement 1A,B*, respectively). The presence GFP-Rab5 on mitochondria was independently confirmed by staining with antibodies against Rab5 and the outer mitochondrial membrane (OMM) protein TOM20 (*Figure 2—figure supplement 1C*).

Next, we examined the kinetics of GFP-Rab5 recruitment to Mito-Red-labeled mitochondria upon laser-induced stress. Time-lapse analysis revealed that the recruitment of Rab5 to rounded mitochondria (*Figure 2C*, Pre- vs Post-laser) was visible within 5 min and reached its peak in signal intensity at ~15 min post-laser treatment (*Figure 2D,E*; data correspond to the inset region of *Figure 2C*). The GFP-Rab5 ring-like signal persisted for >60 min (*Figure 2—figure supplement 2*, arrowheads). In our time-lapse movies, we did not observe evidence of endosomal fusion with mitochondria (*Figure 2E*, *Video 5*), although we cannot exclude fusion with dim vesicles that may have escaped

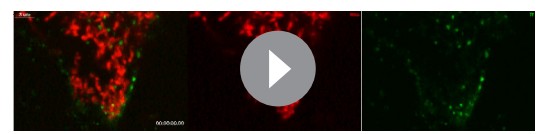

**Video 1.** Dynamics of transferrin and mitochondria. HeLa cells were labeled with Alexa-488 transferrin (Tf; green) and Mito-Red (Mito; red). Images were acquired using a spinning disk confocal microscope at 11 frames/sec for ~6 min. Time stamp corresponds to min: s:ms.

DOI: https://doi.org/10.7554/eLife.32282.005

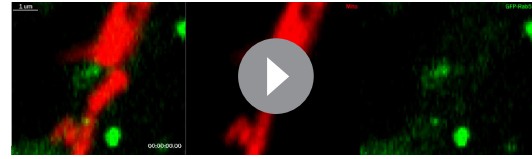

**Video 2.** Dynamics of transferrin and mitochondria. Zoom-in of a HeLa cell labeled with Alexa-488 transferrin (Tf; green) and Mito-Red (Mito; red). The movement of endosomes during a mitochondria fission event is shown. Images were acquired using a spinning disk confocal microscope at 11 frames/sec for ~5 min. Time stamp corresponds to min:s:ms.
DOI: https://doi.org/10.7554/eLife.32282.006

**Video 3.** Dynamics of Rab5-positive endosomes and mitochondria. Zoom-in of a BAC GFP-Rab5 (green) cell labeled with Mito-Red (Mito; red). Images were acquired using a spinning disk confocal microscope at 11 frames/s for ~4 min. Time stamp corresponds to min:s:ms.
DOI: https://doi.org/10.7554/eLife.32282.007

detection. Our interpretation is that the bulk of Rab5 is recruited on mitochondria from the cytosolic pool. These results suggest that upon acute oxidative stress, Rab5 translocates to mitochondria on the time-scale of minutes.

## Rab5 localizes to regions of mitochondria that are stressed

We next asked if GFP-Rab5 translocation to mitochondria is a general response to the overall cell stress or can be elicited locally on individual mitochondria. To test this, we photoirradiated a small area of the cell labeled with Mito-Red (*Figure 2F*, Pre, inset) and monitored the GFP-Rab5 signal after 10 and 20 min. Despite the localized perturbation, the morphological alterations extended to most mitochondria which changed from being tubular to rounded (*Figures 2F*, 10 min). This is consistent with the fact that mitochondria form a dynamically interconnected network (*Lackner, 2014*, *Wang et al., 2015*) that appears to react to local damage as an ensemble. However, the recruitment of Rab5 was limited to only the laser-induced area (*Figures 2F*, 20 min, inset, arrowheads). Similar to *Figure 2A*, distinct GFP-Rab5-positive endosomes contacting mitochondria became visible (*Figure 2F*, inset, double arrowheads). These results suggest that Rab5 is recruited to mitochondria in response to signal(s) originating from individually stressed mitochondria.

## Membrane contacts between Rab5-positive mitochondria and endosomes

The laser-induced stress to either an entire cell or a localized region led to the appearance of Rab5-positive endosomes in close proximity to the swollen mitochondria (*Figure 2A,F*, arrowheads). By live-cell imaging, these endosomes also appeared to dock stably onto mitochondria (*Video 6*, boxed regions, *Video 7*). However, due to the diffraction limit of standard light microscopy, we could not resolve distances that are closer than 200 nm and endosomes that are <200 nm in diameter. To confirm that these are indeed organelle contacts, we complemented our study by correlative light and electron microscopy (CLEM) in order to obtain ultrastructural details. We designed our experimental setup to (1) perform live-imaging of an entire cell, (2) visualize the translocation of GFP-Rab5 and GFP-Rab5-positive endosomes onto mitochondria upon laser-induced stress, and (3) re-image the same cell by serial section transmission electron microscopy (TEM) (*Figure 3A*). GFP-Rab5 cells labeled with Mito-Red were plated on gridded culture dishes, laser-treated and imaged live on the spinning disk confocal microscope. Upon mitochondrial rounding, cells were immediately fixed and re-imaged to assess organellar morphology following fixation. The fluorescent images showed many distinct GFP-Rab5-positive puncta in close proximity to mitochondria (*Figure 3A*). Samples were then processed for serial section TEM, and an inset area tracked by live-cell imaging (*Figure 3B*, red arrowhead) was re-located in the thin sections by both nuclear membrane (*Figure 3B*, dotted line) and mitochondria (*Figure 3B*, red stars) acting as fiducial markers. TEM images from three different serial sections of the inset area revealed that the GFP-Rab5-positive structure corresponds to a tubular-cisternal structure that is approximately 400 nm in diameter, with the typical morphology of an early endosome (*Figure 3C*, green). Serial section analysis showed that the endosomal membrane was in very close contact (<5 nm) with the adjacent mitochondrion of a diameter of ~1.5 μm (*Figure 3C*, red), which showed few cristae. The tomogram showed the presence of membrane contact sites between the rounded mitochondrion and a cisternal structure that could correspond to the

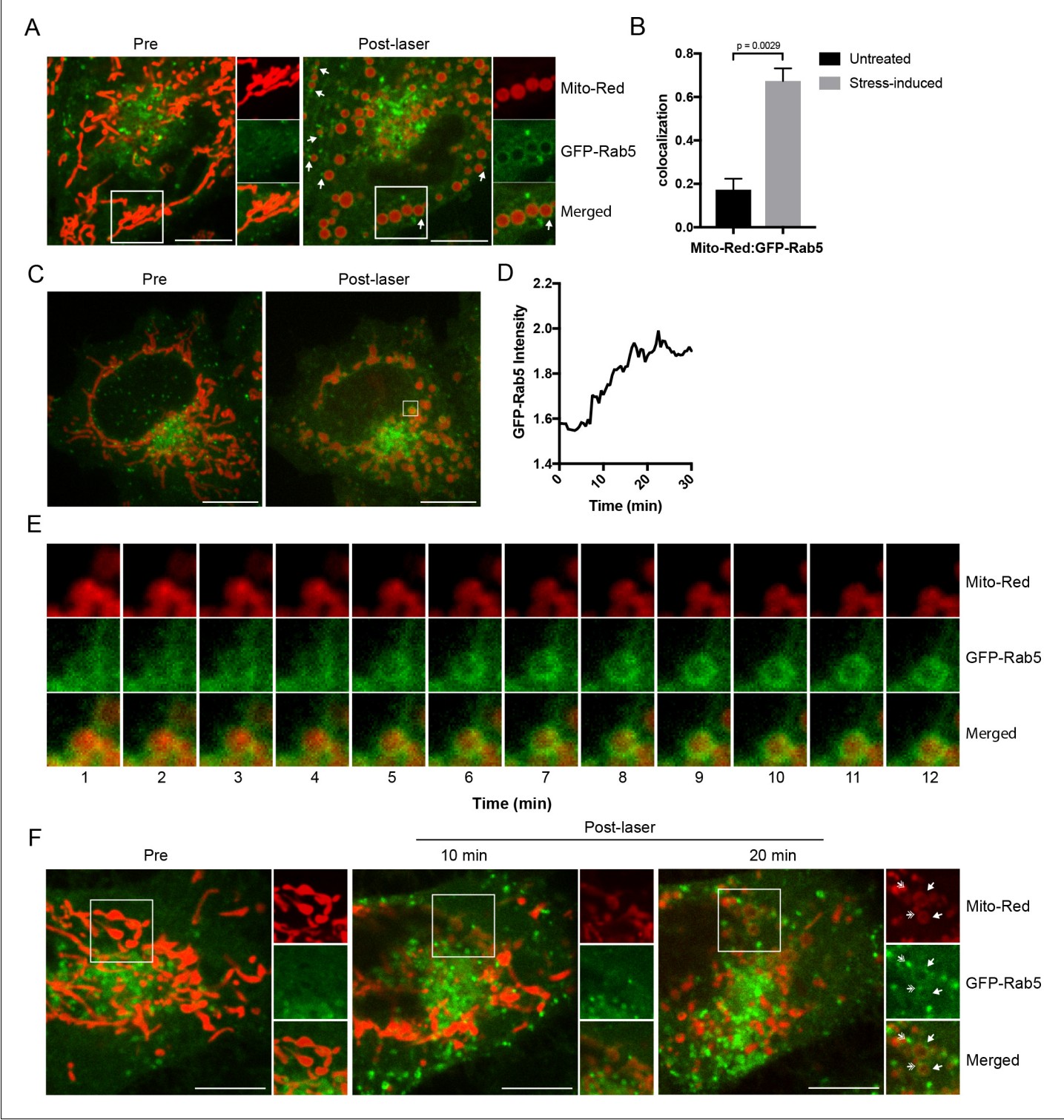

**Figure 2.** Recruitment of Rab5 to mitochondria upon laser-induced stress. (**A**) Live BAC GFP-Rab5 HeLa cells were labeled with 100 nM Mito-Red at 37°C for 30 min. Cells were photoirradiated with a low dosage of 561 nm laser (~5 J/cm$^2$) for 60 s. A snapshot of a cell before (Pre) and 20 min post-laser were taken. Arrowheads indicate GFP-Rab5-positive endosomes that are in close proximity to mitochondria. (**B**) Colocalization analysis between Mito-Red and GFP-Rab5 after photoirradiation as performed in (**A**), *n* = 5. Error bars represent SEMs. p Values based on two-tailed t-tests. (**C**) Live-cell imaging of a BAC GFP-Rab5 HeLa cell treated the same way as in (**A**). (**D**) Quantification of the GFP fluorescent signal over a period of 30 min from the inset region in (**C**). (**E**) Time-lapsed images of Mito-Red, GFP-Rab5, and merged from the inset region in (**C**). (**F**) Live-cell imaging of a BAC GFP-Rab5 HeLa cell photoirradiated only to the marked region (white box). The cell before laser irradiation is shown (Pre). Snapshots of the cell were taken at 10
*Figure 2 continued on next page*

*Figure 2 continued*

min and 20 min post-laser. Inset images show the effect of laser treatment on mitochondrial morphology and network. Arrowheads indicate the recruitment of GFP-Rab5 (filled) and GFP-Rab5-positive endosomes near mitochondria (double-head). Scale bars, 10 µm.

DOI: https://doi.org/10.7554/eLife.32282.008

The following source data and figure supplements are available for figure 2:

**Source data 1.** Numerical data corresponding to the bar graphs presented in Figure B.

DOI: https://doi.org/10.7554/eLife.32282.011

**Source data 2.** Numerical data corresponding to the line trace presented in *Figure 2D*.

DOI: https://doi.org/10.7554/eLife.32282.012

**Figure supplement 1.** Rab5 recruitment to mitochondria is specific to Mito-Red.

DOI: https://doi.org/10.7554/eLife.32282.009

**Figure supplement 2.** GFP-Rab5 enrichment on mitochondria persist for <60 min.

DOI: https://doi.org/10.7554/eLife.32282.010

ER, but did not reveal the presence of double or multiple membranes indicative of mitophagy (*Youle and Narendra, 2011*). Our data suggest that upon mitochondrial stress, two events involving early endosomes and mitochondria occur: (1) Rab5 translocates to mitochondria and (2) endosomes and mitochondria engage in membrane contacts.

## The recruitment of Rab5 to mitochondria is not due to mitophagy

Following mitochondrial stress, both the fast kinetics (min) of Rab5 recruitment on the rounded mitochondria and the absence of wrapped double-membrane structures argue against mitophagy (*Narendra et al., 2008*; *Novak et al., 2010*). We searched for additional evidence to rule out this mechanism. Mitophagy requires the E3 ubiquitin ligase Parkin (*Narendra et al., 2008*), which is normally located in the cytosol but recruited to damaged mitochondria, followed by the formation of LC3-positive autophagosomes and fusion with Lamp1-positive lysosomes in a process that occurs in hours (hr) to days (*Dolman et al., 2013*). To test whether the swollen mitochondria observed in our system undergo this process, we examined the localization of these markers. We first labeled BAC GFP-Rab5 cells with Mito-Red in order to image mitochondria and endosomes at steady state (*Figure 4A,C,E,G*, Untreated) prior to triggering laser-induced stress. Upon fixation, cells were immunostained with specific antibodies to detect endogenous LC3, Lamp1, and Parkin. The cells were seeded on a gridded dish in order to re-locate the same cells following the immunostaining. This allowed us to assess changes to the localization of the markers as a result of stress when compared to neighboring untreated cells (*Figure 4—figure supplement 1*). Cells were then incubated for 60 min at 37°C in order to maximize the time window that these markers might be recruited following GFP-Rab5 translocation to mitochondria. In all laser-induced cells, GFP-Rab5 was specifically enriched around mitochondria when compared to the untreated (*Figure 4B,D,F*). Although we observed a marginal increase in LC3 and Lamp1 colocalization with Mito-Red in stress-induced cells, the signals were mostly concentrated near the perinuclear region (*Figure 4B,D,I*). On the other hand, Parkin remained mostly cytoplasmic

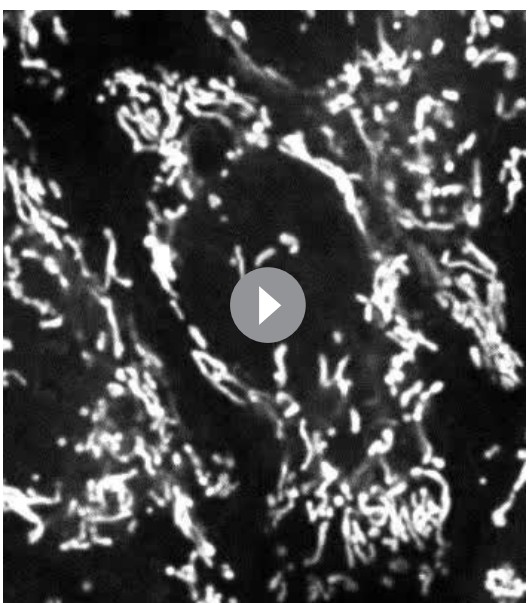

**Video 4.** Mitochondria dynamics during laser-induced stress. HeLa cells were labeled with Mito-Red and photoirradiated via 561 nm laser for 1 min, and immediately imaged. Time-lapse was acquired using a spinning disk confocal microscope at 11 frames/s for ~5 min.

DOI: https://doi.org/10.7554/eLife.32282.013

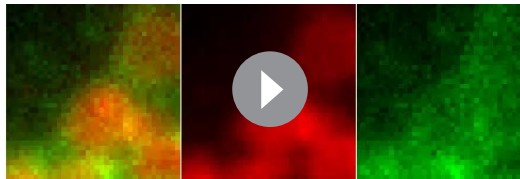

**Video 5.** GFP-Rab5 recruitment to mitochondria during laser-induced stress. Zoom-in of a BAC GFP-Rab5 (green) cell labeled with Mito-Red (Mito; red). Images were acquired using a spinning disk confocal microscope at 11 frames/s with 1 min increment for ~30 min.
DOI: https://doi.org/10.7554/eLife.32282.014

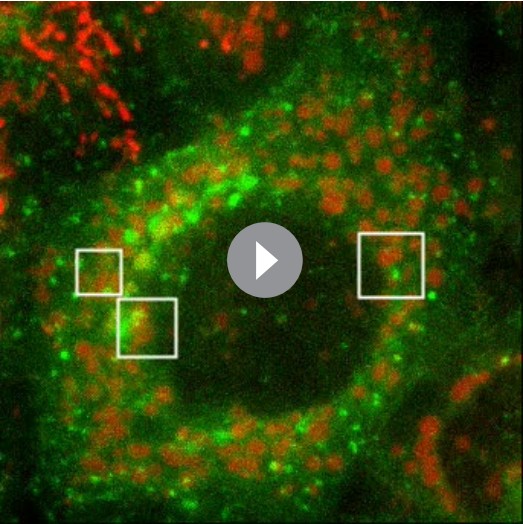

**Video 6.** Endosomes and mitochondria interactions upon laser-induced stress. BAC GFP-Rab5 (green) cells were labeled with Mito-Red (red) and photoirradiated via 561 nm laser for 1 min, and then imaged after 5 min. Time-lapse was acquired using a spinning disk confocal microscope for 20 frames with 5 s increment for ~3 min. Boxed regions indicate GFP-Rab5 endosomes docking onto mitochondria.
DOI: https://doi.org/10.7554/eLife.32282.016

and did not show enrichment around mitochondria in stress-induced cells (*Figure 4F,I*).

As an additional validation of the immunostaining result, we also tested all three markers in HeLa BAC cell lines expressing GFP-tagged LC3, Lamp1, and Parkin by live-cell imaging (*Figure 4—figure supplement 2*). Following laser irradiation, cells were monitored live for 60 min. Similar to the endogenous LC3 and Lamp1 staining, a fraction of the puncta was enriched in the perinuclear region, overlapping with small fragmented mitochondria, but not with the rest of the mitochondria (*Figure 4—figure supplement 2A,B*), suggesting that a low level of autophagy is activated. The GFP-Parkin signals were mostly cytoplasmic in untreated cells but showed a few GFP-Parkin-positive ring-like structures around small fragmented mitochondria, whereas most were devoid of signal in the laser-induced condition (*Figure 4—figure supplement 2C*, arrowheads). Even after 3 hr post-treatment, we observed no significant increase in the number of GFP-Parkin mitochondria (data not shown). The GFP-Parkin ring-like recruitment to these small fragmented mitochondria could be a result of over-expression, inducing some level of mitophagy (*Rana et al., 2013*). Nevertheless, similar to endogenous staining, Parkin was not strongly recruited to the majority of mitochondria. Altogether, the fast kinetics of Rab5 recruitment to stressed mitochondria (<10 min), the absence of a double membranous structure (*Figure 3C*), and the lack of significant Parkin, LC3 and Lamp1 recruitment argue for a mechanism distinct from autophagy and mitophagy.

## Apoptotic signaling is partially activated in laser-induced oxidative stress

The low level of autophagic and mitophagic response raise the question of whether apoptosis is involved. Apoptosis is a programmed cell death pathway which occurs via one of two signaling cascades termed intrinsic and extrinsic pathways (*Tait and Green, 2010*). The intrinsic pathway is initiated through the activation of the Bax/Bak-mediated MOMP, which leads to the release of cytochrome c to activate effector caspases (*Tait and Green, 2010*). Because photoirradiation with Mito-Red engenders oxidative stress in mitochondria (*Hsieh et al., 2015*), we also examined the localization of endogenous Bax via immunostaining. At steady state, Bax

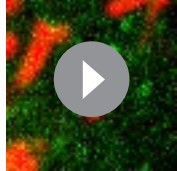

**Video 7.** Endosomes and mitochondria interactions upon laser-induced stress. Zoom-in of a BAC GFP-Rab5 (green) cell labeled with Mito-Red (Mito; red) and photoirradiated via 561 nm laser for 1 min, and then imaged continuously for ~5 min. Images were acquired using a spinning disk confocal microscope at 11 frames/s.
DOI: https://doi.org/10.7554/eLife.32282.017

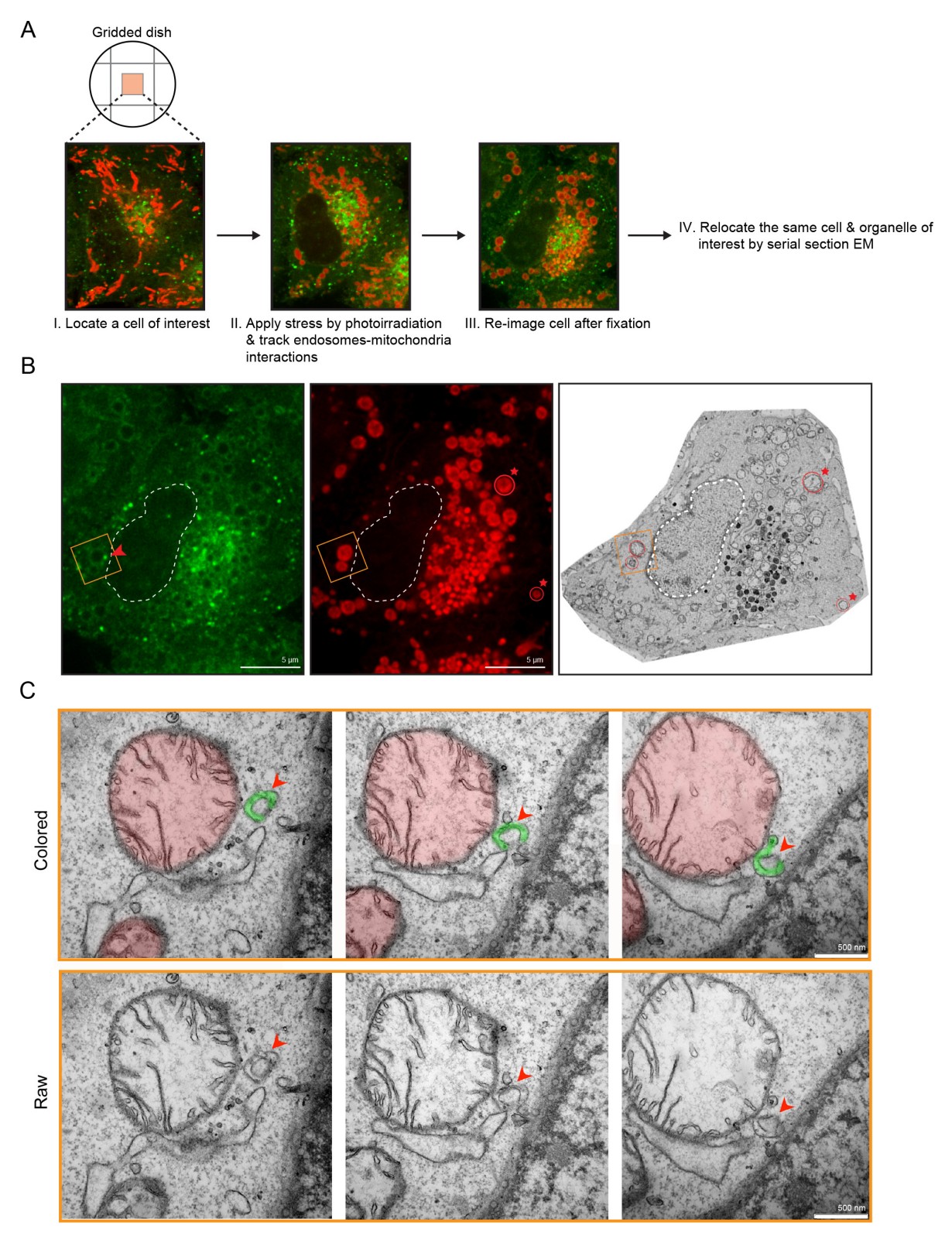

**Figure 3.** Membrane contacts between Rab5-positive mitochondria and endosomes. Ultrastructural analysis of cells upon laser-induced stress. (**A**) Flow chart of the experimental set up. BAC GFP-Rab5 cells were seeded on gridded dishes, labeled with 100 nM Mito-Red at 37°C for 30 min. A cell of interest was then photoirradiated with a low dosage of 561 nm laser (~5 J/cm$^2$) for 60 s. Live-cell imaging between GFP-Rab5-positive endosomes and mitochondria was assessed for organelle dynamics. The same cell was re-imaged for post-fixation analysis before processing for EM. (**B**) Fluorescent

*Figure 3 continued on next page*

*Figure 3 continued*

images of GFP-Rab5 and (Mito-Red) after fixation with glutaraldehyde. Box region shows a GFP-Rab5 endosome (red arrowhead) next to a swollen Rab5-positive mitochondrion. The transmission electron micrograph (TEM) of the same cell is shown (right panel). The nuclear membrane (dotted line) and mitochondria (red star) were used as fiducial markers. (**C**) Zoom-in images of the box region in (**B**). Colored images indicate the GFP-Rab5-positive structure in green and mitochondrion in red. Raw images are shown below. Mitochondrial cristae are visible. The membranous structure adjacent to the mitochondrion likely corresponds to ER. Scale bars, 500 nm.

DOI: https://doi.org/10.7554/eLife.32282.015

signals were visible as cytoplasmic puncta (*Figure 4G*). Upon laser-induced stress, we observed increased Bax puncta around mitochondria compared to the untreated (*Figure 4H,I*). As a positive control for the specificity of the antibody, we treated cells with 10 µM protonophore carbonyl cyanide m-chlorophenyl hydrazone (CCCP), which has been shown to cause Bax translocation to mitochondria (*Mikhailov et al., 2001*; *Saikumar et al., 1998*). Indeed, cells treated with CCCP exhibited higher Bax fluorescence intensity and colocalization with Mito-Red compared to control cells (*Figure 4—figure supplement 3*).

The observed enrichment of Bax on mitochondria upon laser-induced stress led us to ask whether artificially activating apoptosis would also drive Rab5 translocation to mitochondria. One method of triggering mitochondrial-associated apoptosis is the over-expression of the truncated BH3 interacting death domain agonist (tBid), which is a potent inducer of apoptosis by activating Bax and/or self-oligomerization on mitochondria (*Grinberg et al., 2002*). To this end, we infected BAC GFP-Rab5 cells with an adenoviral vector expressing tBid (Ad-tBid) for 12 hr at 37°C followed by TOM20 immunostaining. We found a strong enrichment of GFP-Rab5 on mitochondria only in cells infected with Ad-tBid but not in Ad-control cells (*Figure 4—figure supplement 4*). Our results suggest a mechanism of Rab5 translocation to mitochondria in response to apoptotic signals.

## Hydrogen peroxide triggers Rab5 translocation to mitochondria and affects mitochondrial respiration without disrupting membrane potential

What could be the signal that drives Rab5 recruitment? Several possible scenarios such as morphological changes to mitochondria and/or release of mitochondrial-derived factor(s) may account for this. Morphological changes such as matrix condensation or swelling of mitochondria are often associated with MOMP by Bax activation, cytochrome c release and subsequent activation of caspases (*Gottlieb et al., 2003*). However, this is not a prerequisite. For example, the mitochondrial uncoupler CCCP causes mitochondrial swelling and rounding but without immediate cytochrome c release or cell death (*Gao et al., 2001*; *Lim et al., 2001*). On the other hand, hydrogen peroxide ($H_2O_2$) induces mitochondrial rounding associated with increased Bax expression (*Gutiérrez-Venegas et al., 2015*), cytochrome c release and caspase activation (*Gutiérrez-Venegas et al., 2015*; *Takeyama et al., 2002*). For these reasons, we first investigated the effect of CCCP and $H_2O_2$ on Rab5 localization. Whereas mitochondria were mostly tubular in control cells (*Figure 5A,B*, top panels), the exposure of cells to either CCCP or $H_2O_2$ for 2 hr resulted in mitochondrial rounding and fragmentation (*Figure 5A,B*, bottom panels), as previously reported (*Narendra et al., 2008*; *Pletjushkina et al., 2006*). Interestingly, CCCP did not cause Rab5 enrichment on mitochondria, which was only observed in $H_2O_2$-treated cells (*Figure 5A,B*, arrowheads), quantified as colocalization with Mito-Red (*Figure 5C,D*). Consistent with previous findings, we found that the release of cytochrome c into the cytosol was upregulated in $H_2O_2$-treated, but not in CCCP-treated cells (*Figure 5E*). Since cytochrome c is a known factor for activating caspase-dependent programmed cell death, we also assessed the activity of caspase 3/7 via a 4-amino acid peptide (DEVD) conjugated to a DNA-binding dye. Cleavage of the DEVD peptide by caspase 3/7 releases the DNA-binding fragment, yielding a fluorescent signal. Using flow cytometry, we detected ~62% of cells showing strong fluorescent signal in $H_2O_2$-treated cells, and merely ~0.6% and~6.2% in control and CCCP-treated cells, respectively. As additional controls of the treatment, we showed that only CCCP, but not $H_2O_2$, resulted in the recruitment of GFP-Parkin as well as endogenous Parkin on mitochondria (*Figure 5—figure supplement 1A,B*).

Given the primary role of mitochondria in cellular metabolism, we asked whether the observed differences between CCCP and $H_2O_2$ could be related to altered mitochondrial respiration. To test

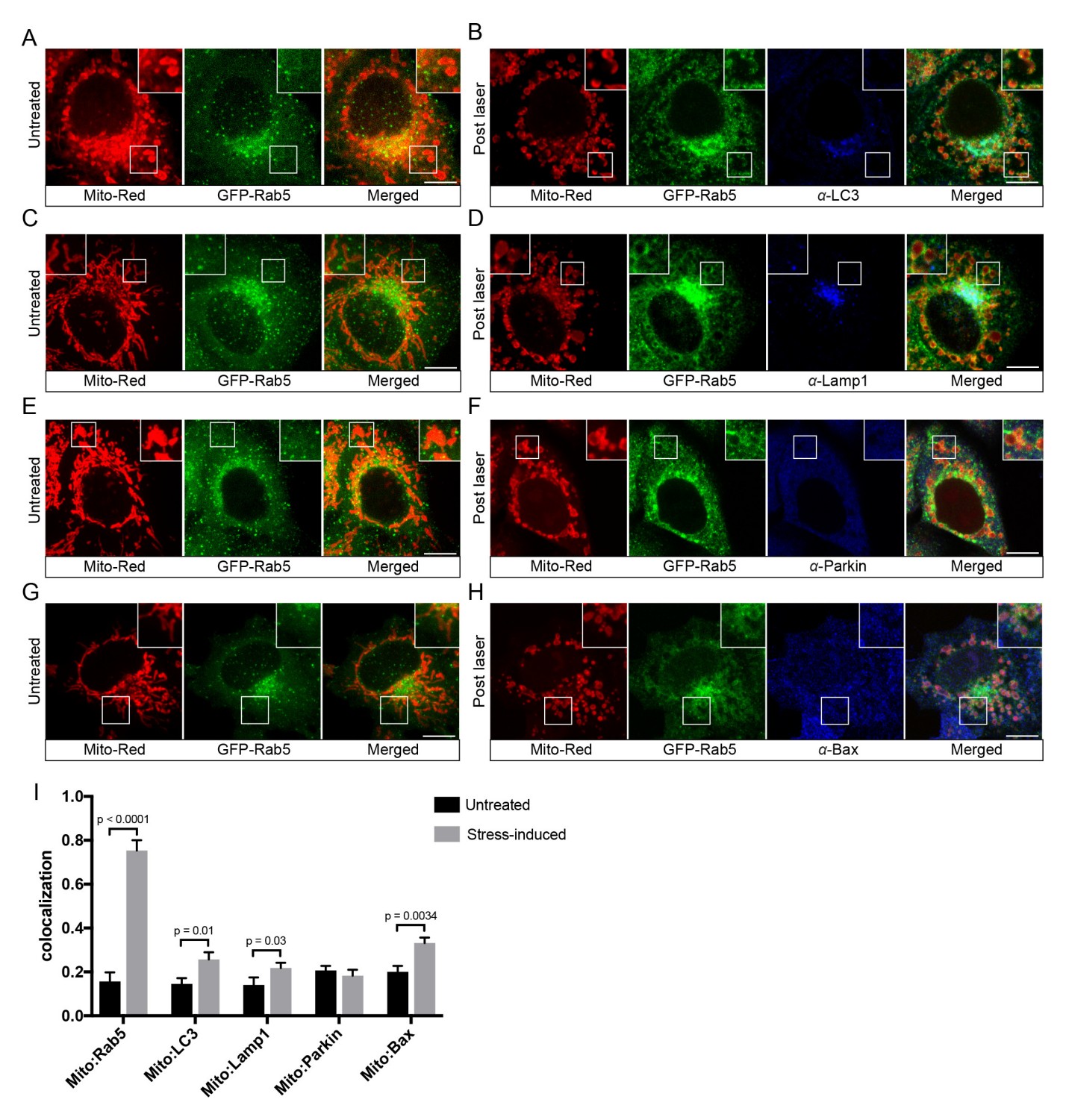

**Figure 4.** Localization of endogenous LC3, Lamp1, Parkin, and Bax upon laser-induced stress. BAC GFP-Rab5 cells were labeled with 100 nM Mito-Red at 37°C for 30 min. Cells were acquired prior to laser irradiation as controls (Untreated) (**A**), (**C**), (**E**), and (**G**). Same cells were photoirradiated as before, fixed after 60 min post-laser treatment, and immunostained with specific antibodies against LC3 (**B**), Lamp1 (**D**), Parkin (**F**), and Bax (**H**). Inset images show the area before and after laser treatment. (**I**) Colocalization analysis of Mito-Red to Rab5, LC3, Lamp1, Parkin, and Bax, in untreated vs laser-induced conditions (*n* = 3). Untreated cells correspond to cells outside of the laser treated area. Error bars represent SEMs. p Values based on two-tailed t-tests. Scale bars, 10 μm.

DOI: https://doi.org/10.7554/eLife.32282.018

*Figure 4 continued on next page*

*Figure 4 continued*

The following source data and figure supplements are available for figure 4:

**Source data 1.** Numerical data corresponding to the bar graphs presented in *Figure 4I*.

DOI: https://doi.org/10.7554/eLife.32282.024

**Figure supplement 1.** Localization of endogenous LC3, Lamp1, and Parkin in untreated cells.

DOI: https://doi.org/10.7554/eLife.32282.019

**Figure supplement 2.** BAC GFP -LC3, -Lamp1, and -Parkin localization upon laser-induced stress.

DOI: https://doi.org/10.7554/eLife.32282.020

**Figure supplement 3.** CCCP *induces* Bax protein expression and localization on mitochondria.

DOI: https://doi.org/10.7554/eLife.32282.021

**Figure supplement 3—source data 1.** Numerical data corresponding to the bar graphs presented in *Figure 4—figure supplement 3B,C*.

DOI: https://doi.org/10.7554/eLife.32282.022

**Figure supplement 4.** tBid over-expression leads to Rab5 enrichment on mitochondria.

DOI: https://doi.org/10.7554/eLife.32282.023

this, we measured the oxygen consumption rate (OCR) in live HeLa cells grown in the presence of galactose in order to force cells to rely primarily on oxidative phosphorylation as opposed to glycolysis (*Aguer et al., 2011*). Upon the addition of CCCP, a sharp increase in OCR was recorded compared to the media control (*Figure 5G*, purple vs orange curve). This is because CCCP causes the collapse of the proton gradient and the disruption of the mitochondrial membrane potential. As a result, electrons flow unhindered through the electron transport chain, boosting the OCR. In contrast, injection of $H_2O_2$ resulted in an initially sharp decline in OCR, but quickly returned to its earlier steady state within 20 min (*Figure 5G*, green curve), suggesting that the effect on OCR is reversible. The decrease in respiration rate by $H_2O_2$ was similarly reported in intact cardiac mitochondria, in which α-ketoglutarate was used as a respiratory substrate (*Nulton-Persson and Szweda, 2001*). Altogether, our data show that, in addition to activating caspase-dependent apoptotic program, $H_2O_2$ also induces the translocation of Rab5 to mitochondria without disrupting the membrane potential.

## Hydrogen-peroxide-induced stress triggers Rab5 translocation from EE to mitochondria, increases early endosomal-mitochondrial contacts, and interferes with transferrin uptake

Given the key role of Rab5 in the biogenesis of the endosomal system (*Zeigerer et al., 2012*; *Zerial and McBride, 2001*), the translocation of Rab5 to mitochondria upon oxidative stress by $H_2O_2$ led us to investigate the connection between endocytosis and oxidative stress. Our live-cell imaging data (*Figures 2* and *3*) argue that GFP-Rab5 is not delivered to mitochondria by fusion with EE. Rather, it may be released from EE and recruited to mitochondria via a cytosolic intermediate. To test this, we collected the total membrane (M) and cytosolic (C) fractions by subcellular fractionation from HeLa cells treated with/out $H_2O_2$ and immunoblotted for Rab5, the endosomal tether Rab5 effector EEA1, GAPDH (as cytosolic marker), and TOM20 (a mitochondrial marker) (*Figure 6A*). It was previously shown that $H_2O_2$-induced stress causes increased Rab5-GDI complex in BHK cells (*Cavalli et al., 2001*). Consistent with this, upon 1 hr $H_2O_2$ treatment, we detected a ~ 0.5-fold increase in the amount of Rab5 in the cytosolic fraction compared to the control (*Figure 6A*, lane 2,4), supporting the view that the cytosolic pool increases at the expense of the membrane-associated pool. The cytosolic levels of Rab5 then decreased at 2 hr time point (*Figure 6A*, lane 4,6, *Figure 6B*). We also detected an increase in the EEA1 pool to the cytosol in the long exposure upon $H_2O_2$ treatment (*Figure 6A*, right blot). Since Rab5 in the M fraction includes both the EE and mitochondrial pool, we specifically measured the EE-associated Rab5 by estimating the colocalization of Rab5 with EEA1 by confocal immunofluorescence microscopy. We found a marked (~35%) decrease in the colocalization in $H_2O_2$-treated compared to untreated cells (*Figure 6C*). Altogether, our results suggest that oxidative stress induces the solubilization of a fraction of Rab5 from the endosomal membrane into the cytosol and its translocation to the mitochondrial membrane.

The partial dissociation of Rab5 and EEA1 from the endosomal membrane suggests that endocytic trafficking may be affected. To test this, we stimulated HeLa cells with Alexa-647 Tfn

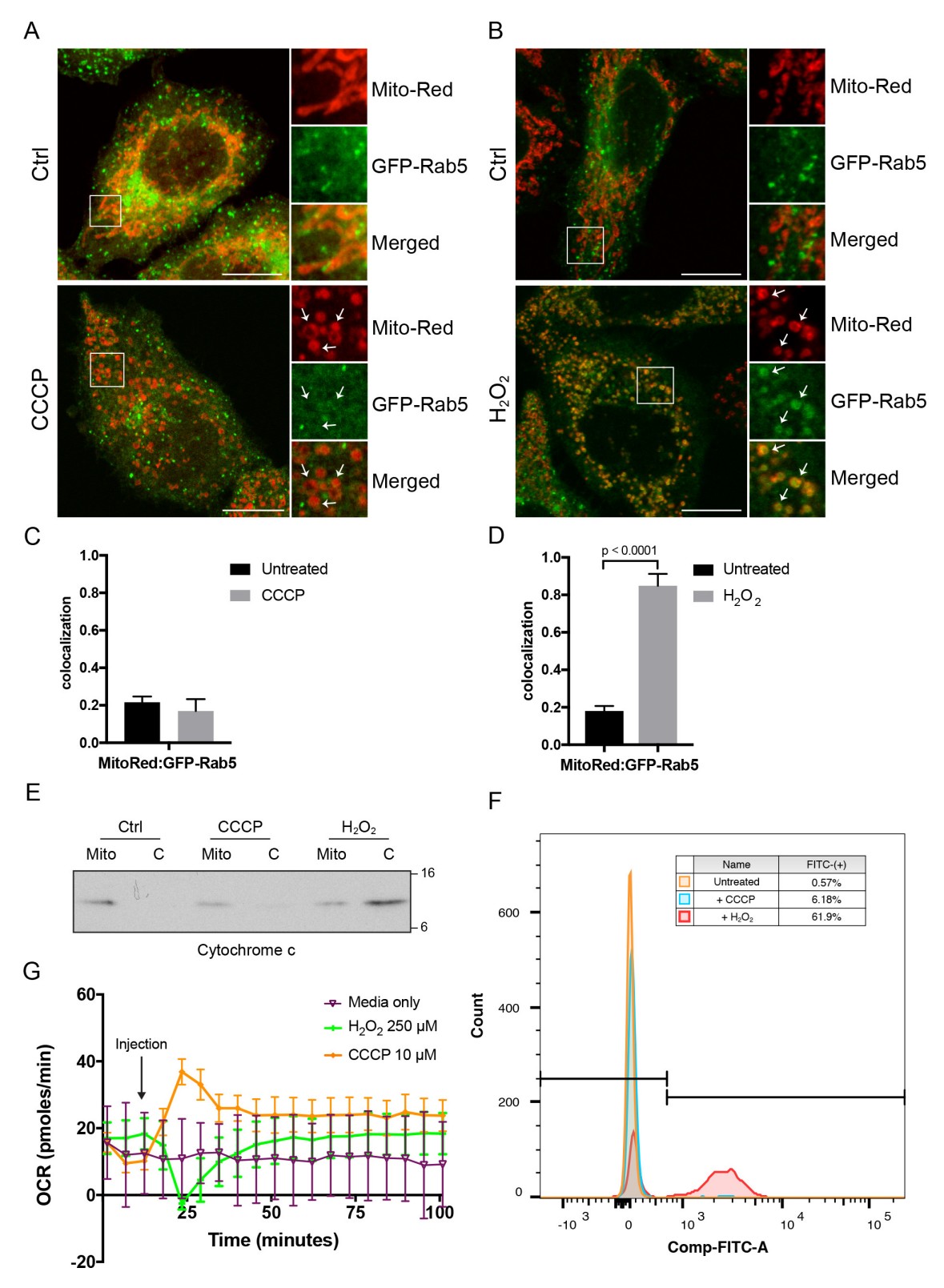

**Figure 5.** Effects of *CCCP vs H$_2$O$_2$* on Rab5 recruitment to mitochondria. BAC GFP-Rab5 HeLa cells labeled with 100 nM of Mito-Red were treated with either DMSO (Ctrl), 10 μM CCCP (**A**), or 250 μM H$_2$O$_2$ at 37°C for 2 hr (**B**). Cells were fixed and imaged by confocal microscopy. Inset regions reveal the effect on mitochondrial morphology and GFP-Rab5 localization upon treatment. Arrowheads indicate rounded and stressed mitochondria in both CCCP- and H$_2$O$_2$- treated conditions. Scale bars, 10 μm. (**C**) and (**D**) Colocalization analysis between Mito-Red and GFP-Rab5 in (**A**) and (**B**),

*Figure 5 continued on next page*

*Figure 5 continued*

respectively; *n* = 50. Error bars represent SEMs. p Values based on two-tailed t-tests. (**E**) Subcellular fractionation was performed in HeLa cells treated with DMSO (Ctrl), 10 μM CCCP, or 250 μM $H_2O_2$ at 37°C for 2 hr. Protein samples from purified mitochondria (Mito) and cytosolic (**C**) fractions were loaded onto SDS-PAGE and imunoblotted with cytochrome c antibody. (**F**) Cells were treated the same way as in (**E**). Cells were then resuspended in live-cell imaging solution containing 500 nM caspase-3/7 green flow cytometric reagent and incubated at 37°C for 30 min before subjecting to flow cytometric analysis. FITC signal (x-axis) is plotted against the total cell count (y-axis). The gating was set based on the background signal in DMSO control. (**G**) HeLa cells were seeded in the Seahorse 96-well plate format and incubated overnight. Growth medium was exchanged to bicarbonated-free low-buffered assay medium (provided by the manufacturer) supplemented with 10 mM galactose immediately before the start of the experiment. Oxygen consumption rate measurements were measured in the Seahorse XFe96 Analyzer from cells injected with media only (purple), 250 μM $H_2O_2$ (green), or 10 μM CCCP (orange). Error bars represent SEMs.

DOI: https://doi.org/10.7554/eLife.32282.025

The following source data and figure supplement are available for figure 5:

**Source data 1.** Numerical data corresponding to the bar graphs presented in *Figure 5C,D*.
DOI: https://doi.org/10.7554/eLife.32282.027
**Source data 2.** Numerical data corresponding to the line traces presented in *Figure 5G*.
DOI: https://doi.org/10.7554/eLife.32282.028
**Figure supplement 1.** *CCCP, but not $H_2O_2$, leads to Parkin recruitment.*
DOI: https://doi.org/10.7554/eLife.32282.026

continuously for 5 min at 37°C. In the absence of $H_2O_2$, transferrin was efficiently internalized into endosomes (*Figure 6D*, Untreated). In contrast, cells pre-treated with $H_2O_2$ for 10 min showed a severe block in Tfn uptake with an accumulation of Tfn signal on the cell surface (*Figure 6D,E*), confirming the inhibition of endocytic uptake.

In the laser-induced stress conditions, we observed a high occurrence of EE contacting stressed mitochondria (*Figure 2A,F*,*Figure 3*), and these interactions appeared to be specific to EE (*Figure 1*). The effects by $H_2O_2$ prompted us to further probe whether differential interactions exist between mitochondria and Tfn- vs. EGF-positive endosomes. To test this, HeLa cells stained with MitoRed were continuously labeled with either Tfn-488 or EGF-488 for 10 min at 37°C and then incubated for 50 min with/out $H_2O_2$. There was a significant increase in the colocalization between Mito-Red and both Tfn-488 and EGF-488 in $H_2O_2$-treated compared to non-treated cells (*Figure 1C,D*). Intriguingly, we also found an increase in the total signal intensity in $H_2O_2$ conditions (*Figure 1E*), in which Tfn signals accumulated more prominently in the perinuclear region (typical of RE), whereas EGF was more evenly distributed compared to untreated (*Figure 1C*). Increased colocalization with $H_2O_2$ is likely attributed to a block in recycling of Tfn (as uptake is inhibited) and sorting of EGF to LE/lysosome, thereby resulting in cargo accumulation in EE.

Altogether, our data suggest that oxidative stress induced by laser irradiation or $H_2O_2$ leads to Rab5 translocation from EE to mitochondria, increased EE-mitochondrial MCS, and a defect in endosomal sorting.

## Stress-induced Rab5 translocation to mitochondria blocks cytochrome c release

By live-cell imaging, we found that mitochondria respond to $H_2O_2$ treatment with different kinetics within a cell (*Video 8*). Distinct regions of the mitochondrial network were more prone to rounding and membrane permeabilization than others, as revealed by the differential loss of Mito-Red during $H_2O_2$ treatment (*Figure 7A*, inset image). Regions containing stressed rounded mitochondria correlated exclusively with the Rab5 ring-like recruitment (*Figure 7A*, inset image, arrowheads), suggesting that Rab5 may be involved in either facilitating or preventing the apoptotic process. Therefore, we asked whether Rab5 plays a role in regulating cytochrome c release. For these experiments, we transiently over-expressed GFP-Rab5 (or GFP as control) in HeLa cells and measured the amount of cytosolic cytochrome c at different time points after incubation with $H_2O_2$. We found a significant delay in the cytochrome c release from mitochondria in GFP-Rab5 over-expressing cells compared to control cells (*Figure 7B,C*). Our results suggest that Rab5 plays a protective role in mitochondrial-induced apoptosis by down-regulating the release of pro-apoptotic factor(s) to the cytosol.

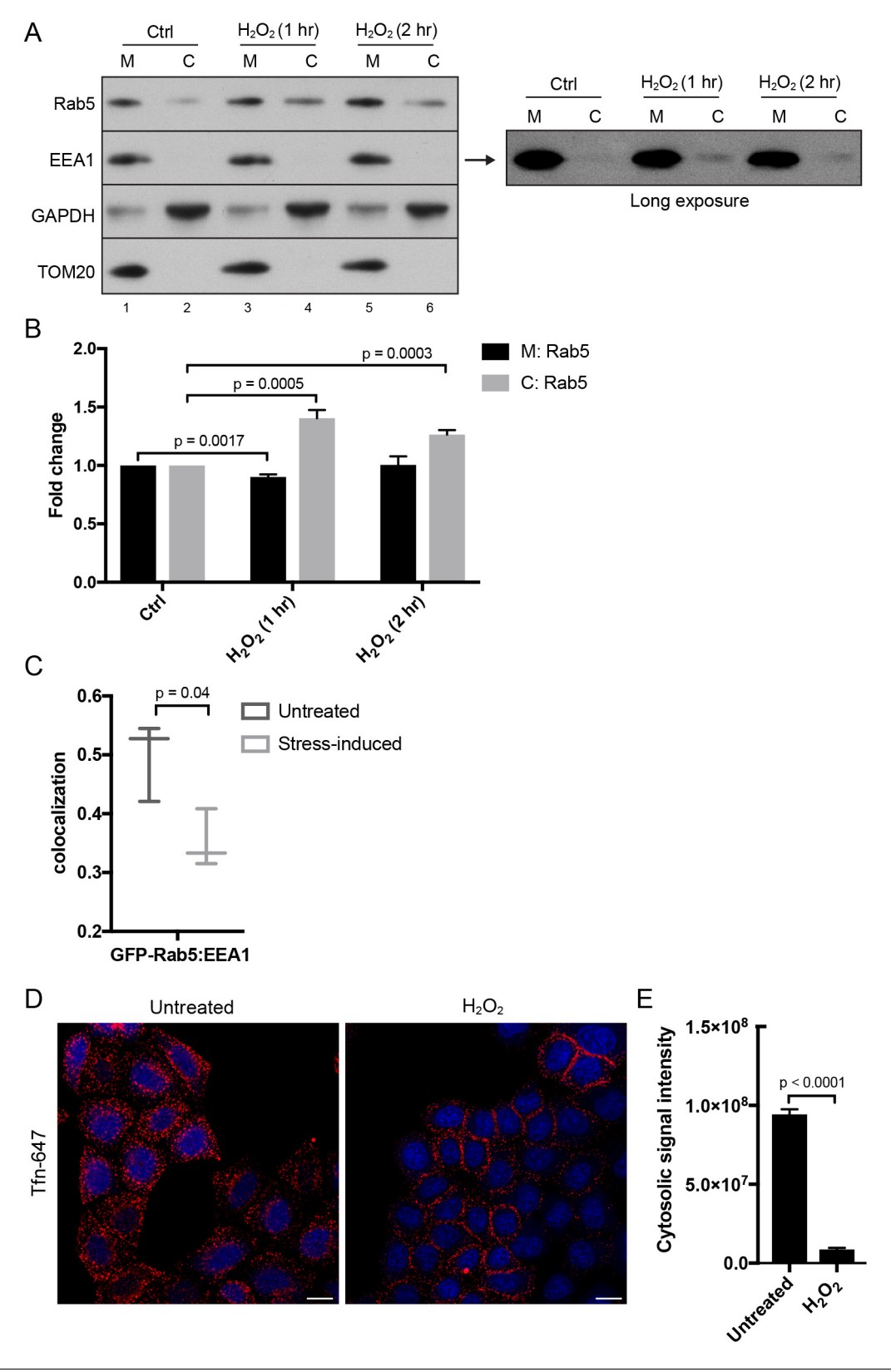

**Figure 6.** *Hydrogen peroxide (H₂O₂) treatment decreases* Rab5 membrane association on EE, increases early endosomal-mitochondrial contacts, and reduces transferrin uptake. (**A**) Subcellular fractionation was performed in
*Figure 6 continued on next page*

*Figure 6 continued*

HeLa cells treated with $H_2O_2$ for 1 and 2 hr. The total membrane (**M**) fraction was obtained by centrifugation of the post nuclear supernatant at 200,000 g at 4°C for 1 hr, and supernatant was taken as cytosolic (**C**) fraction. Protein samples were loaded onto SDS-PAGE and imunoblotted with antibodies against Rab5, EEA1, GAPDH, and TOM20. The long exposure blot for EEA1 is shown (right panel). (**B**) Densitometric quantification of Rab5 in (**A**). Band intensities were calculated as normalized ratio between Rab5 to TOM20 in the M fraction, and Rab5 to GAPDH in the C fraction. Fold change is plotted on the y-axis. Error bars represent SEMs from three independent experiments. (**C**) BAC GFP-Rab5 HeLa cells were incubated with/out 250 µM $H_2O_2$ *at* 37°C for 2 hr. Cells were fixed and immunostained with EEA1 antibody. Colocalization analysis was performed between GFP-Rab5 and EEA1. (**D**) HeLa cells were seeded in a 384-well plate and pre-treated with either PBS (control) or 250 µM $H_2O_2$ at 37°C for 10 min. Cells were then pulsed with Alexa-647 Tfn (10 µg/ml) for 5 min, washed with PBS, fixed, and stained with DAPI (nuclear) and CellMask Blue (cytoplasmic) dyes. Images were acquired by the Yogokawa confocal microscope. Scale bars, 10 µm. (**E**) Quantification of the cytoplasmic fluorescence intensity per cell, *n* = 50. p Values based on two-tailed t-tests.

DOI: https://doi.org/10.7554/eLife.32282.029

The following source data is available for figure 6:

**Source data 1.** Numerical data corresponding to the bar graph presented in *Figure 6B*.
DOI: https://doi.org/10.7554/eLife.32282.030

**Source data 2.** Numerical data corresponding to the chart presented in *Figure 6C*.
DOI: https://doi.org/10.7554/eLife.32282.031

**Source data 3.** Numerical data corresponding to the bar graphs presented in *Figure 6E*.
DOI: https://doi.org/10.7554/eLife.32282.032

## Rab5 enrichment on mitochondria is reversible

We reasoned that if the translocation of Rab5 to mitochondria induced by oxidative stress is a pro-survival response by lowering the apoptotic potential, then the process should be reversible when stress is removed and mitochondria may recover their normal state. The fast recovery rate of mitochondrial respiration following $H_2O_2$ injection (*Figure 5E*) also supports this prediction. We initially tested a range of $H_2O_2$ concentrations (100 µM to 1 mM) and incubation times (2 to 24 hr), in order to find an optimal balance between a measurable level of Rab5 translocation to mitochondria and minimal cell death. We found that concentrations up to 500 µM did not cause a noticeable cell rounding by 2 hr (data not shown). Therefore, we pre-incubated cells with 250 µM $H_2O_2$ over a period of 24 hr and quantitatively measured TOM20 levels and Rab5-mitochondria colocalization as a means to assess mitochondrial mass and interaction dynamics. Cells were incubated in the presence or absence of $H_2O_2$ for 6, 12 and 24 hr, and either lysed for western blot analysis or fixed for immunostaining. Interestingly, we found that the levels of TOM20 started to increase after 6 hr, suggesting that the mitochondrial mass increased (*Figure 8A*, left panels). In immunostained cells, we found that the colocalization between endogenous TOM20 and Rab5 (*Figure 8B*) peaked at 6 hr and started to taper off after 12 hr (*Figure 8C*). Decreased colocalization correlated with increased TOM20 protein levels (*Figure 8A*). Remarkably, the removal of $H_2O_2$ at 12 hr followed by an additional 12 hr incubation in complete medium not only fully restored the morphology of mitochondria from rounded to tubular (*Figure 8B*, bottom panel), but also returned TOM20:Rab5

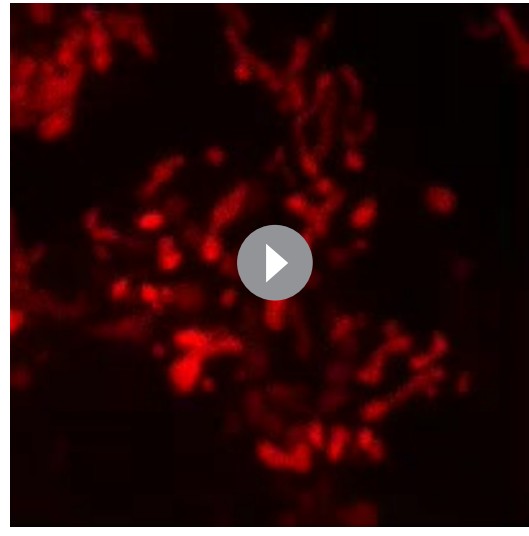

**Video 8.** Mitochondria dynamics during $H_2O_2$-induced stress. HeLa cells were labeled with Mito-Red and then imaged in the presence of 250 µM $H_2O_2$ for ~60 min. Time-lapse was acquired using a spinning disk confocal microscope for 50 frames with 2-min increment.
DOI: https://doi.org/10.7554/eLife.32282.035

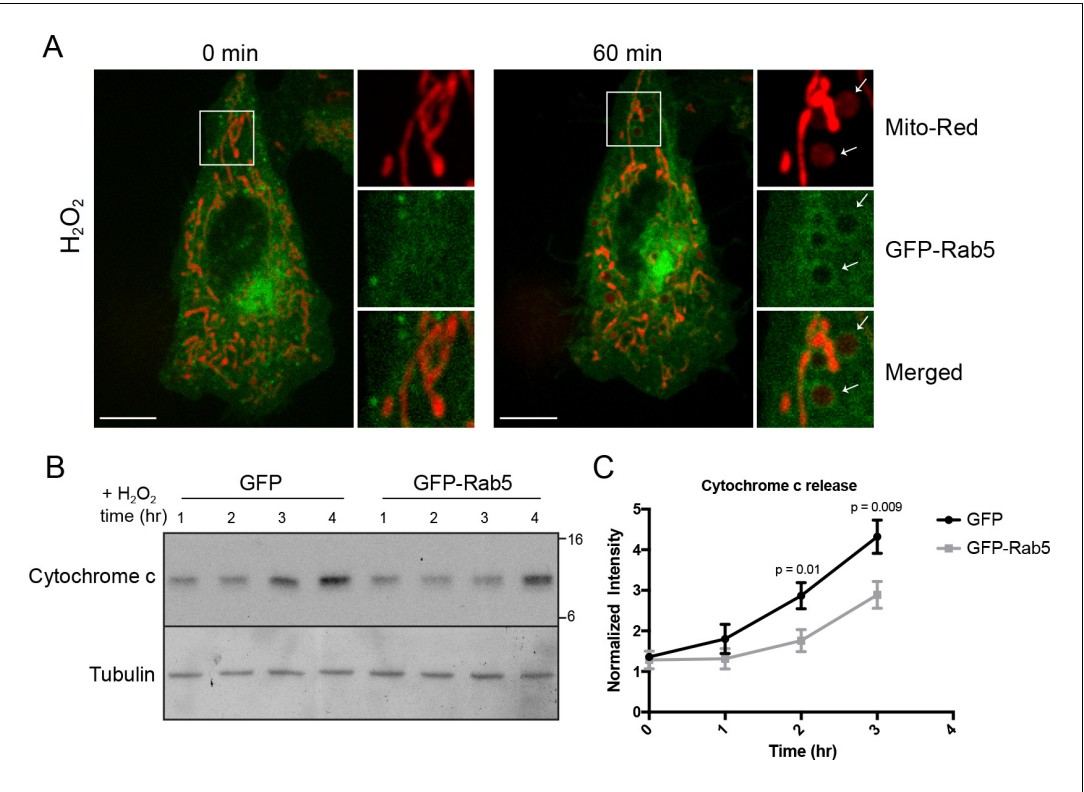

**Figure 7.** Rab5 regulates cytochrome c release during $H_2O_2$-induced stress. (**A**) Live-cell imaging of a BAC GFP-Rab5 cell stained with 100 nM Mito-Red before (0 min) and after treatment (60 min) with 250 µM $H_2O_2$ at 37°C. Inset images are shown and arrowheads indicate GFP-Rab5 recruitment to MOMP events (right panel). Scale bars, 10 µm. (**B**) HeLa cells over-expressing either GFP or GFP-Rab5 were treated with 250 µM $H_2O_2$ at 37°C for 1 to 4 hr. Protein samples from cytosolic fractions were obtained and immunoblotted for cytochrome c and tubulin (as a loading control). (**C**) Densitometric quantification of cytochrome c release in (**B**). Data were collected from three independent experiments. Y-axis corresponds to the normalized ratio intensity of cytochrome c to the tubulin loading control. p Values based on two-tailed t-tests.

DOI: https://doi.org/10.7554/eLife.32282.033

The following source data is available for figure 7:

**Source data 1.** Numerical data corresponding to the line traces presented in *Figure 7C*.
DOI: https://doi.org/10.7554/eLife.32282.034

colocalization to steady state levels (*Figure 8C*). The response was dose-dependent as cells exposed to 500 µM $H_2O_2$ showed arrested TOM20 levels and were unable to be rescued despite $H_2O_2$ removal at 12 hr (*Figure 8A*, right panels). Our data show that Rab5 translocation to mitochondria induced by oxidative stress is a reversible and protective process responding to apoptotic signals via the regulation of mitochondria.

## Rab5 enrichment on mitochondria is accompanied by specific effector recruitment

Because Rab5 translocates from EE to mitochondria with a consequent reduction in endocytic uptake, we next asked whether the endosomal Rab5 effectors are also recruited to mitochondria. We systematically assessed the localization of various endosomal effectors such as Rabenosyn-5, EEA1, APPL1 and APPL2 in BAC GFP-Rab5 HeLa cells labeled with Mito-Red via immunostaining by pair-wise combinations. We deliberately chose to detect the endogenous because the tagged proteins often severely perturb the endosomal system, as assessed by quantitative endocytic trafficking (*Kalaidzidis et al., 2015*). Specific antibodies were first tested in untreated control cells, which showed significant levels of colocalization with GFP-Rab5 (*Figure 9—figure supplement 1*). Upon

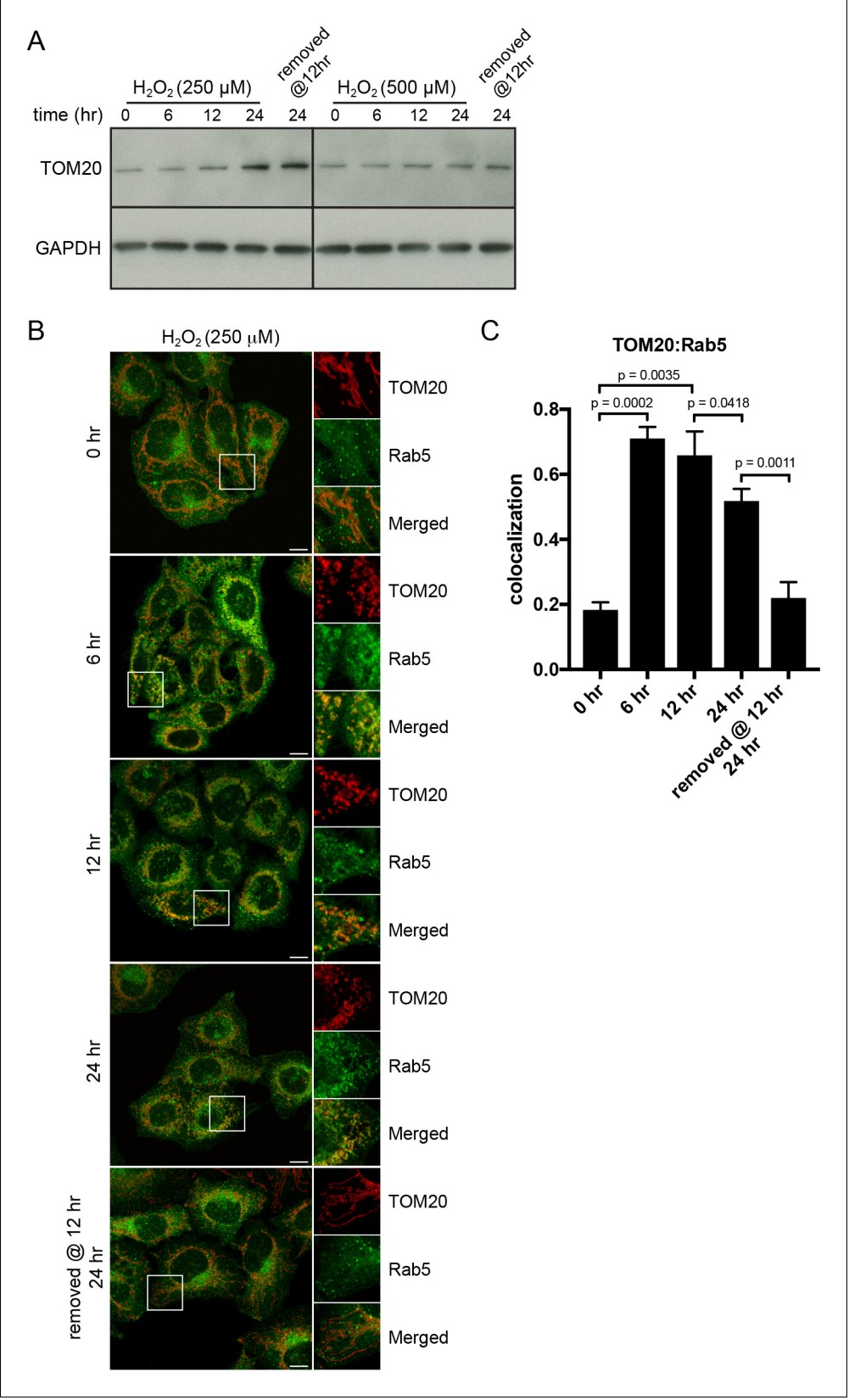

**Figure 8.** $H_2O_2$-induced translocation of Rab5 to mitochondria is reversible. HeLa cells were seeded in 24-well plates (**A**) or on glass coverslips (**B**), and after overnight incubation, were treated with either 250 µM or 500 µM $H_2O_2$ at 37°C for 0, 6, 12, 24 hr, or replaced with a fresh medium at 12 hr followed by additional 12 hr incubation. *Figure 8 continued on next page*

*Figure 8 continued*

(**A**) Cells were then harvested in SDS loading buffer and protein lysates were loaded onto SDS-PAGE and immunoblotted with TOM20 and GAPDH antibodies. (**B**) Cells were fixed and immunostained with Rab5 and TOM20 antibodies. Inset regions are shown. Scale bars, 10 µm. (**C**) Colocalization analysis between TOM20 and Rab5 in (**B**), *n* = 50. p Values based on two-tailed t-tests.

DOI: https://doi.org/10.7554/eLife.32282.036

The following source data is available for figure 8:

**Source data 1.** Numerical data corresponding to the bar graphs presented in *Figure 8C*.

DOI: https://doi.org/10.7554/eLife.32282.037

laser-induced stress, the appearance of GFP-Rab5 rings around mitochondria provided an immediate visual cue and served as a positive control. Cells were fixed after 30-min incubation post-laser treatment. Of the tested effectors, we detected a strong enrichment of Rabenosyn-5, but not EEA1, on mitochondria in the same cell (*Figure 9A,C*). Neither APPL1 nor APPL2 showed enrichment around mitochondria, despite a robust Rab5 recruitment (*Figure 9B,C*). Unlike Rabenosyn-5, EEA1, APPL1 and APPL2 remained well distributed in endosomal-like vesicles in both treated and untreated cells (*Figure 9A,B*, *Figure 9—figure supplement 1B,C,D*). As Rabenosyn-5 and EEA1 are recruited to endosomes via both Rab5 and PI(3)P-binding FYVE motifs (*Nielsen et al., 2000*), we asked whether phosphatidylinositol 3-phosphate (PI(3)P) was present on mitochondria in our stress conditions. To test this, we over-expressed the PI(3)P probe GFP-2xFYVE$^{Hrs}$ (*Gillooly et al., 2000*) in HeLa cells and monitored GFP signals in live cells upon laser-induced stress. Fluorescence signals were present as vesicle-like puncta (*Figure 9—figure supplements 2*, 0 min), as previously reported (*Gillooly et al., 2000*). After 60 min, stressed and swollen mitochondria were observed, but these were completely devoid of GFP signals, which remained on vesicle-like puncta (*Figure 9—figure supplements 2*, 60 min).

To corroborate the immunostaining data with an independent method, we also tested the effect of $H_2O_2$ on Rab5 and Rab5 effectors in association with mitochondria by subcellular fractionation. We isolated cytosolic (C) and mitochondrial fractions (Mito) via differential centrifugation and probed them with different effector antibodies by western blot (*Figure 9D*). Consistent with our immunostaining results, Rab5 and Rabenosyn-5, but not EEA1 and APPL1/2, were found to be specifically enriched in the mitochondrial fraction (marked by TOM20) treated with $H_2O_2$ compared to non-treated (*Figure 9D*, lane 2,4).

Altogether, our findings reveal a selective mechanism of Rab5 translocation and activation on mitochondria in the absence of PI(3)P.

## The Rab5 GEF Alsin localizes to mitochondria upon stress induction

Translocation and recruitment of effectors imply that Rab5 must be activated on the mitochondrial membrane. Activation of Rab GTPases on organelle membranes depends on a family of GEFs (*Blümer et al., 2013*; *Pfeffer, 2013*; *Zerial and McBride, 2001*; *Zhen and Stenmark, 2015*). We first examined the localization of Rabex-5, a known GEF of Rab5 on the endosomal membrane, by immunostaining in BAC GFP-Rab5 cells. For reasons described above, we visualized the endogenous protein because tagged Rabex-5 constructs proved not to be functional as judged by their perturbations on the endosomal system (Kalaidzidis and Zerial, unpublished data). The formation of GFP-Rab5 rings upon laser treatment served as a positive control. Despite a modest enrichment on mitochondria upon laser-induced stress, endogenous Rabex-5 remained mostly cytosolic and on cytoplasmic puncta (*Figure 10A*), consistent with its endosomal localization (*Figure 10—figure supplement 1A*).

The localization pattern of Rabex-5 led us to hypothesize that another GEF might be principally involved. We turned our attention to Alsin based on several lines of evidence. Alsin is the gene product of ALS2, which is mutated in multiple neurodegenerative disorders such as juvenile amyotrophic lateral sclerosis (ALS), juvenile primary lateral sclerosis (JPLS), and infantile-onset ascending hereditary spastic paralysis (IAHSP). Alsin comprises three GEF domains: (1) a RCC1-like GEF domain for Ran GTPase, (2) a DH-PH domain for Rho GTPase, and (3) a C-terminal VPS9 domain for Rab5 (*Topp et al., 2005*) (*Figure 10—figure supplement 1B*). Functional studies in ALS mouse models

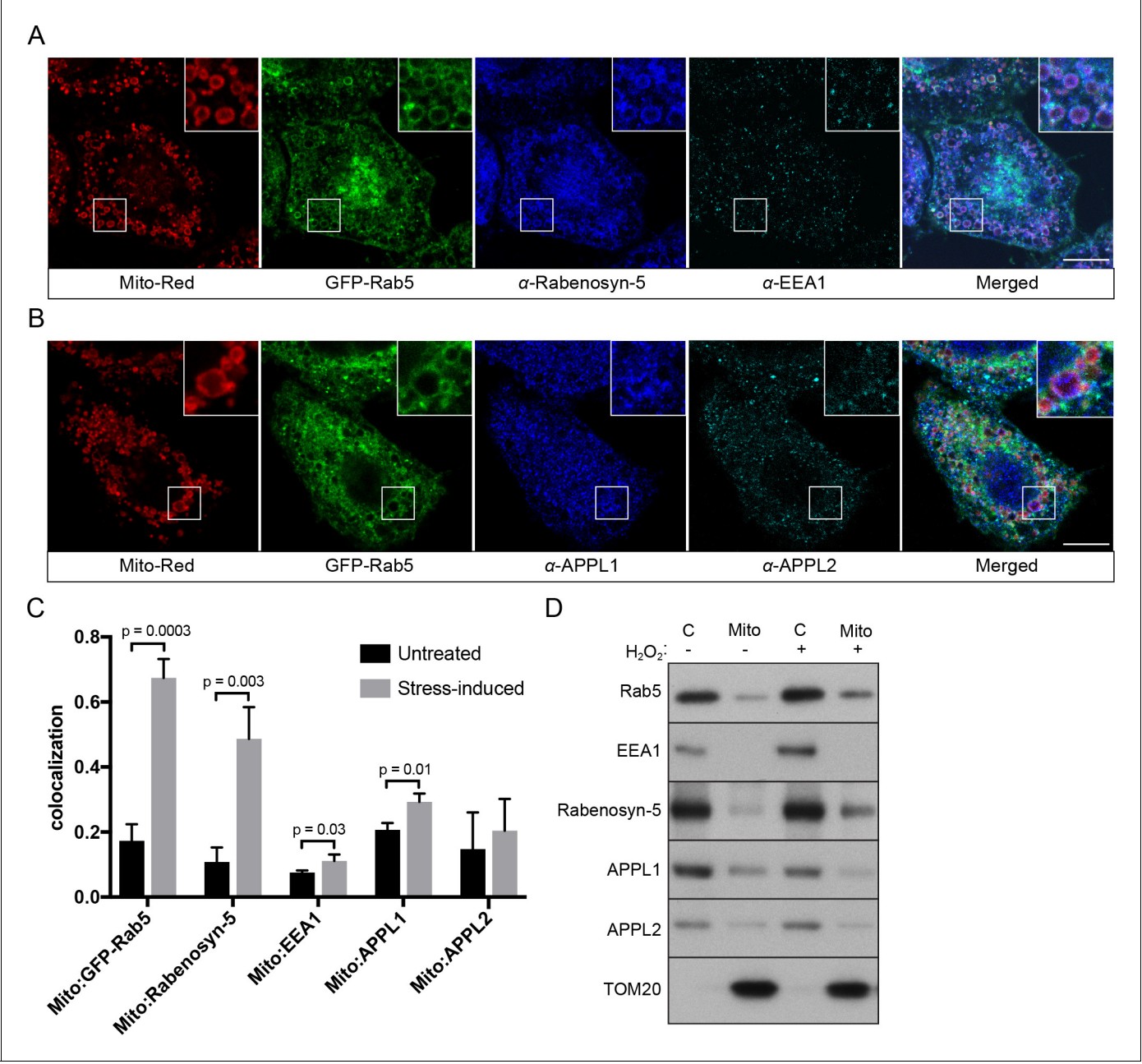

**Figure 9.** Localization of Rab5 effectors upon mitochondrial stress. BAC GFP-Rab5 cells seeded on gridded dishes were labeled with Mito-Red and photoirradiated as before. Cells were fixed after 30 min post-laser treatment and immunostained with antibodies against ZFYVE20 and EEA1 (**A**), or APPL1 and APPL2 (**B**). Inset regions are shown. Scale bars, 10 μm. (**C**) Colocalization analysis from untreated and laser-treated cells in (**A**) and (**B**), n = 3. p Values based on two-tailed t-tests. (**D**) Subcellular fractionation was performed in HeLa cells treated with either PBS (control) or 250 μM $H_2O_2$ at 37°C for 2 hr. Protein samples from purified mitochondria (Mito) and cytosolic (**C**) fractions were loaded onto SDS-PAGE and imunoblotted with antibodies against Rab5, EEA1, Rabenosyn-5, APPL1, APPL2, and TOM20.

DOI: https://doi.org/10.7554/eLife.32282.038

The following source data and figure supplements are available for figure 9:

**Source data 1.** Numerical data corresponding to the bar graphs presented in *Figure 9C*.
DOI: https://doi.org/10.7554/eLife.32282.041

**Figure supplement 1.** Localization of endogenous Rabenosyn-5, EEA1, APPL1 and APPL2.
DOI: https://doi.org/10.7554/eLife.32282.039

**Figure supplement 2.** Localization of GFP-2xFYVE[Hrs] upon laser-induced stress.
DOI: https://doi.org/10.7554/eLife.32282.040

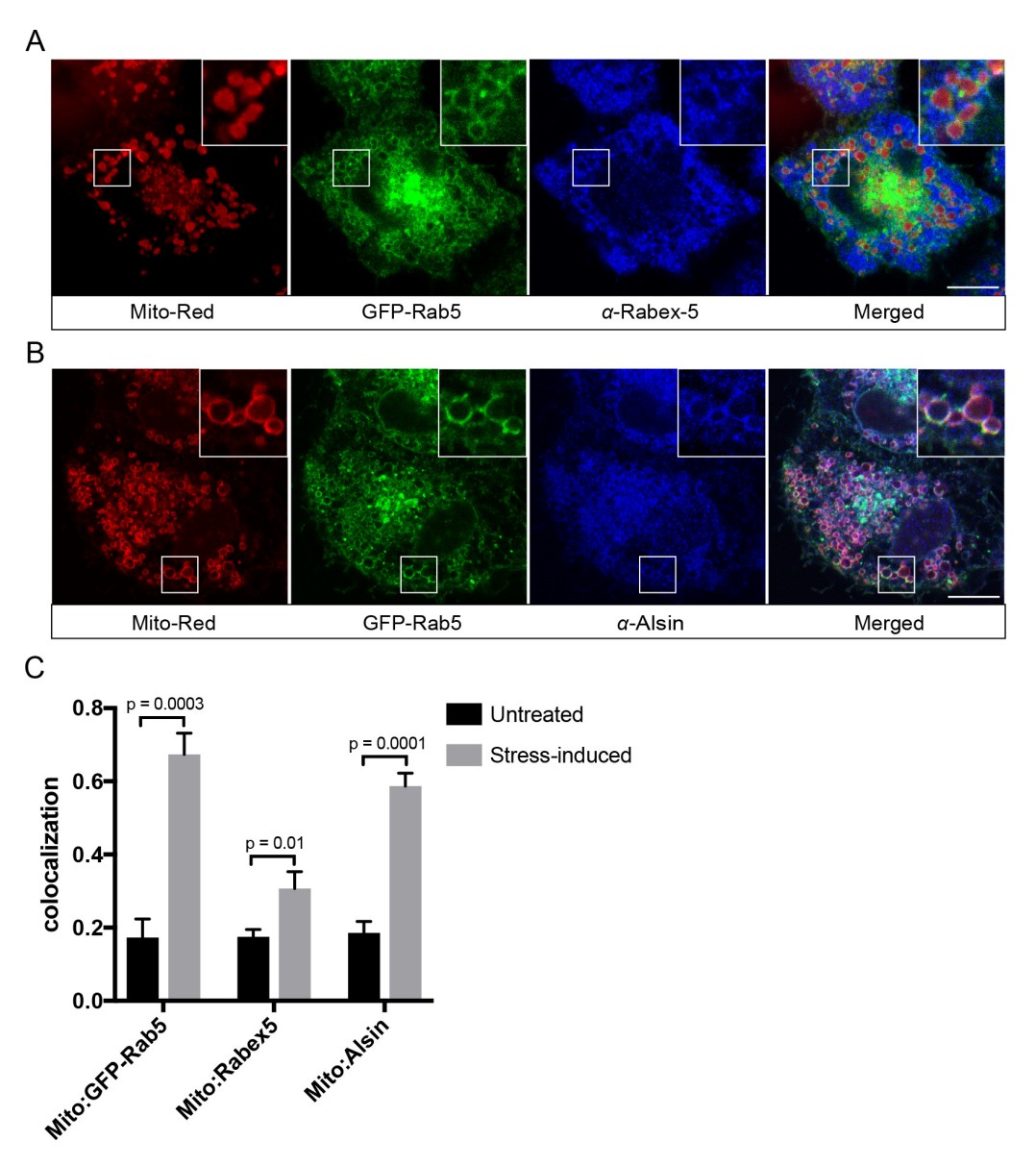

**Figure 10.** Localization of Rabex-5 and Alsin upon mitochondrial stress. BAC GFP-Rab5 cells seeded on gridded dishes were labeled with 100 nM Mito-Red and photoirradiated as before. Cells were fixed after 30 min post-laser treatment and immunostained with antibody against Rabex-5 (**A**) or Alsin (**B**). Inset regions are shown. Scale bars, 10 μm. (**C**) Colocalization analysis from untreated and laser-treated cells in (**A**) and (**B**), *n* = 3. p Values based on two-tailed t-tests.

DOI: https://doi.org/10.7554/eLife.32282.042

The following source data and figure supplements are available for figure 10:

**Source data 1.** Numerical data corresponding to the bar graphs presented in *Figure 10C*.
DOI: https://doi.org/10.7554/eLife.32282.046

**Figure supplement 1.** Endogenous localization of Rabex-5 and Alsin.
DOI: https://doi.org/10.7554/eLife.32282.043

**Figure supplement 2.** Over-expression of Alsin or Rab5 blocks caspase-3/7 activation.
DOI: https://doi.org/10.7554/eLife.32282.044

**Figure supplement 2—source data 1.** Numerical data corresponding to the bar graphs presented in *Figure 10—figure supplement 2B*.
DOI: https://doi.org/10.7554/eLife.32282.045

have associated Alsin with neuronal survival (*Kanekura et al., 2004*; *Panzeri et al., 2006*) and endo-lysosomal trafficking (*Hadano et al., 2016*; *Hadano et al., 2010*). Moreover, corticospinal motor neuron (CSMN) in Alsin KO mice display selective defects in mitochondrial morphology (*Gautam et al., 2016*). At steady state, Alsin localized to vesicular structures, showing partial overlap with Rab5 (*Figure 10—figure supplement 1C*), consistent with the reported localization of Alsin (*Kanekura et al., 2004*; *Topp et al., 2004*). However, after laser treatment, Alsin exhibited a strong and uniform staining around mitochondria (*Figure 10B*), where it showed significant colocalization with GFP-Rab5 and Mito-Red (*Figure 10C*).

The spatial and functional connection between Alsin and Rab5 suggest that Alsin may also be implicated in stress-induced response on mitochondria. We tested this idea by over-expressing either Alsin or WT Rab5 in HeLa cells and found that both prevented caspase 3/7 activation as revealed by the weak fluorescence signals (due to the lack of cleavage on the DEVD-conjugated DNA-binding dye) compared to control cells, when challenged with $H_2O_2$ (*Figure 10—figure supplement 2A,B*). Our results point to Alsin as a candidate GEF for activating Rab5 on mitochondria upon stress induction.

## Alsin regulates mitochondrial apoptotic signaling and is required for efficient Rab5 targeting to mitochondria

Several mouse models have been generated for the studies on Alsin. However, these models have failed to recapitulate the phenotypes observed in human patients (*Cai et al., 2008*). It has recently been reported that the absence of Alsin appears to specifically affect the health of corticospinal motor neurons (*Gautam et al., 2016*). Therefore, in order to directly probe the role of Alsin in a more physiological background without compromising our ability for genetic and chemical manipulations, we generated Alsin CRISPR knockout cells in human-induced pluripotent stem cells (iPSCs). We confirmed the deletion of the Alsin gene by sequencing (not shown) and RT-PCR, and the encoded protein by western blot (*Figure 11—figure supplement 1A,B*). We were then able to differentiate both WT and mutant (Alsin$^{-/-}$) iPSCs into spinal motor neurons (iPSC-sMNs) using a previously reported protocol (*Reinhardt et al., 2013*). In short, we induced neural progenitor cells (NPC) through embryonic bodies formation by growing iPSC in a medium supplemented with transforming growth factor-ß (TGF- ß) and bone morphogen protein (BMP) small molecule inhibitors (SB431542 and dorsomorphin, respectively), and WNT and Sonic Hedgehog signaling activators (CHIR99021 and PMA, respectively). Differentiation and maturation stages were achieved by culturing cells in retinoic acid (RA), cAMP, and neurotrophic factors (BDNF and GDNF) (*Figure 11A*). As a quality control, high expression of pluripotency markers such as Oct4 and Lin28 were observed in our iPSCs as well as Nestin, Sox2 and Pax6 expression in our neuro-progenitor cells (NPCs) (*Figure 11—figure supplement 1C*). Differentiation into mature spinal motor neurons was validated by the expression of choline acetyltransferase (ChAT), HB9, and Islet-1 (ISL1) (*Figure 11—figure supplement 1D,E*). These cells showed extensive axonal network as revealed by the MAP2 staining (*Figure 11—figure supplement 1E*). Finally, mature spinal motor neurons were re-stained for the expression of Alsin in both WT and Alsin$^{-/-}$ cells (*Figure 11—figure supplement 1F*).

We first examined the steady-state localization of Rab5 and morphology of mitochondria by immunostaining for endogenous Rab5 and TOM20. Similar to hippocampal neurons (*de Hoop et al., 1994*), Rab5 was ubiquitously present on endosomal-like vesicles in the soma, dendrites and axon in iPSC-sMNs. The mitochondrial network in iPSC-sMNs was less tubular and contained more numerous and smaller rounded mitochondria than those in HeLa cells (*Figure 11B*, Ctrl). Next, we tested whether iPSC-sMNs would exhibit the same mitochondrial response to oxidative stress as observed in HeLa cells. Noticeably, we found iPSC-sMNs to be more susceptible to detachment and cell rounding than HeLa cells when challenged with 250 μM $H_2O_2$ for 2 hr under the same conditions (data not shown). We thus lowered the $H_2O_2$ concentration to 100 μM such that no immediate cell detachment nor rounding were observed during the treatment. We then examined the morphology of mitochondria, the association of Rab5 with mitochondria, and the release of cytochrome c into the cytosol. At steady state, we did not observe significant alterations in mitochondria morphology in both WT and Alsin$^{-/-}$ iPSC-sMNs. However, WT iPSC-sMNs challenged with $H_2O_2$ showed a robust enrichment of Rab5 on mitochondria, but not in Alsin$^{-/-}$ iPSC-sMNs (*Figure 11B*, $H_2O_2$).

To corroborate these results, we also performed subcellular fractionation of iPSC-sMNs. In control cells, endogenous Rab5 was detected primarily in the cytosolic fraction and minimally in the

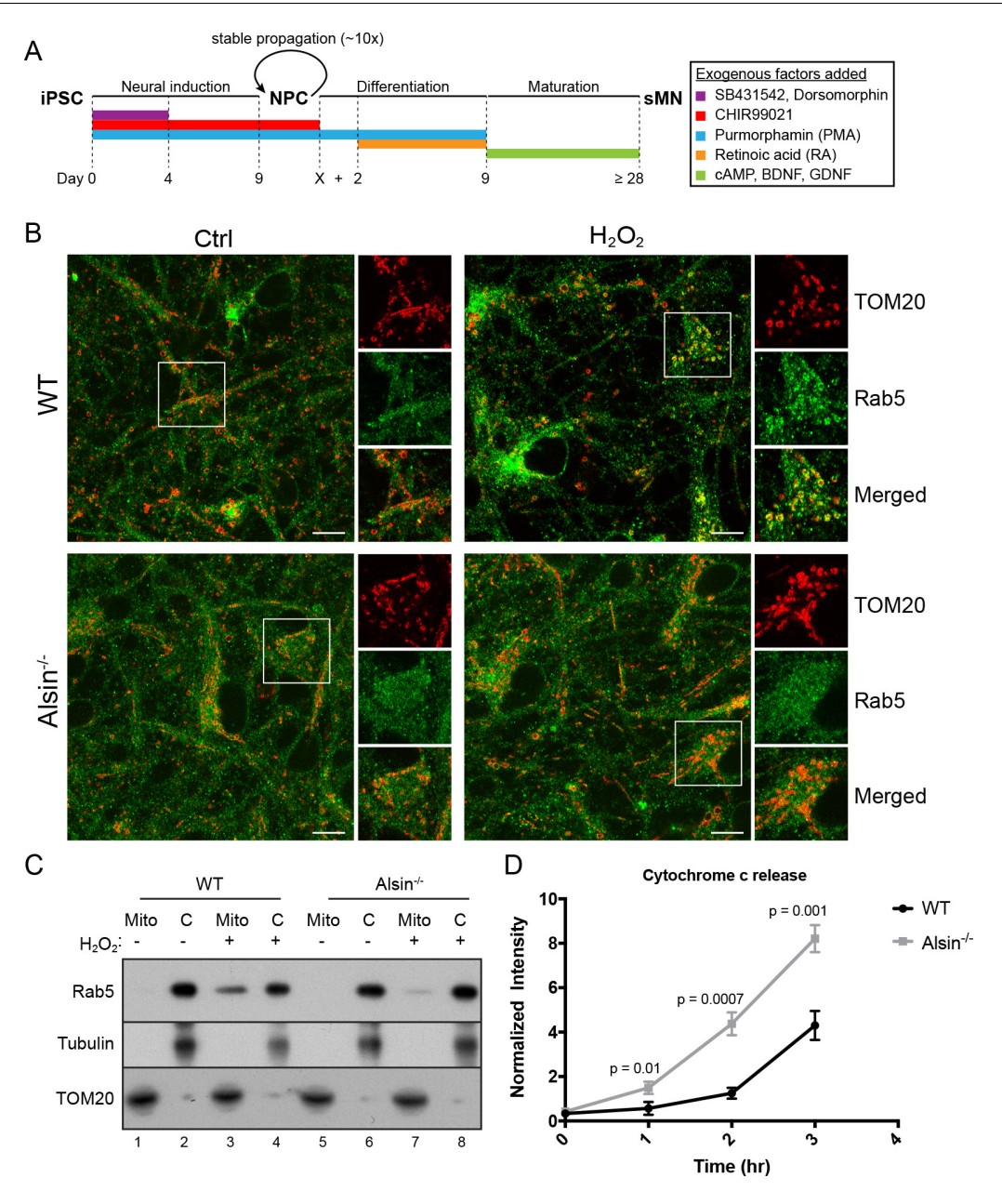

**Figure 11.** Alsin is required for Rab5 recruitment and regulates cytochrome c release. (**A**) Flow chart depicting the different stages and time (in days) from induced-pluripotent stem cells (iPSC), to neuroprogenitor cells (NPC), and to mature spinal motor neurons (sMN). The small molecules and compounds used at different stages are shown and color-coded. (**B**) WT and Alsin[-/-] cells were challenged with either PBS (Ctrl) or 100 µM $H_2O_2$ at 37°C for 1 hr. Cells were fixed and immunostained with Rab5 and TOM20 antibodies. Inset images show are shown. Scale bars, 10 µm. (**C**) Subcellular fractionation of mitochondrial (Mito) and cytosolic (C) fractions from WT and Alsin[-/-] iPSC-sMN challenged with either PBS or 100 µM $H_2O_2$ at 37°C for 1 hr. Protein samples were loaded onto SDS-PAGE and imunoblotted with antibodies against Rab5, tubulin and TOM20. (**D**) Cytosolic fractions were prepared from WT and Alsin[-/-] iPSC-sMN challenged with 100 µM $H_2O_2$ at 37°C for 1 hr. Densitometric quantification of cytosolic cytochrome c were collected from three independent experiments. Y-axis corresponds to normalized ratio intensity of cytochrome c to tubulin. Error bars represent SEMs. p Values based on two-tailed t-tests.
DOI: https://doi.org/10.7554/eLife.32282.047

The following source data and figure supplement are available for figure 11:

**Source data 1.** Numerical data corresponding to the line traces presented in *Figure 11D*.

*Figure 11 continued on next page*

*Figure 11 continued*

DOI: https://doi.org/10.7554/eLife.32282.049

**Figure supplement 1.** Validation of the CRISPR/Cas9 Alsin<sup>-/-</sup> cells in iPSC, NPC and iPSC-sMN.

DOI: https://doi.org/10.7554/eLife.32282.048

mitochondrial fraction (*Figure 11C*, lane 1,2,5,6). On the other hand, cells challenged with $H_2O_2$ showed a strong enrichment of Rab5 co-fractionating with the mitochondrial fraction in WT iPSC-sMNs, but only weakly in Alsin$^{-/-}$ iPSC-sMNs (*Figure 11C*, lane 3,7). The lack of Rab5 enrichment in Alsin$^{-/-}$ cells also correlated with a greater susceptibility to $H_2O_2$-induced apoptotic signaling, as assessed by the rapid release of cytochrome c into the cytosol within 1 hr and subsequent accumulation, when compared to WT cells (*Figure 11D*). Collectively, our findings demonstrate that Alsin is a key regulator for recruiting Rab5 to mitochondria, which altogether, impart a cytoprotective function for cells against oxidative stress.

## Discussion

We discovered a novel cytoprotective mechanism during oxidative stress entailing the translocation of Rab5 from EE to mitochondria. Interestingly, the activation of Rab5 requires Alsin, which has been implicated in early onset ALS. Our results provide an unexpected mechanistic link between the endosomal system and mitochondria that could be of primary importance for understanding the mechanistic cause of ALS and other neurodegenerative diseases.

Different nutrient or environmental perturbations can affect mitochondria morphology and metabolic activities such as oxidative phosphorylation and programmed cell death (*Galloway and Yoon, 2013*). Mitochondria can elicit adaptive responses to oxidative stress that may lead to hypoxia adaptation, inflammation, or programmed cell death (*Sena and Chandel, 2012*). Our findings suggest that the endocytic system is a primary responder to mitochondria under oxidative stress. Laser- or exogenous ROS (e.g. $H_2O_2$)-induced damage causes MOMP, mitochondrial swelling, and release of cytochrome c, leading to caspase activation and apoptosis. Under these conditions, the endosomal system appears to rapidly respond to damaged mitochondria through a rescue pathway, which results in the recruitment of Alsin and Rab5 to mitochondria, inhibition of cytochrome c release, decrease in mitochondrial oxygen consumption and hence, increased overall cell viability (*Figure 12*). In the course of this study, a mitochondrial clearance mechanism was reported where Rab5-positive EE sequester mitochondria via the ESCRT machinery when cells are treated with the proton uncoupler FCCP (*Hammerling et al., 2017*), an analog of CCCP. Our mechanism appears to be distinct from this as well as the canonical autophagic/mitophagic mechanisms. First, we did not observe the engulfment of mitochondria into Rab5-positive EE but rather, the recruitment of Alsin, Rab5, and Rabenosyn-5 on mitochondria, as well as an increase in early endosomal-mitochondrial MCS in response to stress. This is also distinct from the intra-mitochondrial recruitment of Rab5 and endolysosomes upon over-expression of the apoptotic factors (*Hamacher-Brady et al., 2014*). Second, we did not observe engulfing membraneous structures around stressed mitochondria nor upon CCCP treatment. One explanation could be the use of different cell types and the lower concentration of CCCP employed in our experiments. Third, the recruitment of Rab5 to damaged mitochondria occurs rapidly, that is within min, well preceding any autophagic components that we analyzed in this study. We found that autophagy is restricted to only a subset of small mitochondrial fragments that are LC3$^+$, whereas the majority are devoid of known autophagic markers such as Parkin, LC3 and Lamp1. We could not rule out that mitochondrial clearance mechanism may still be activated at a later time. However, our data with $H_2O_2$ show that the mechanism described here is reversible (*Figure 8*) and argue for a mitochondrial-protective role rather than a degradative process. In fact, the reversal and recovery of cells from late-stage apoptosis (i.e. following cytochrome c release and caspase activation) have recently been reported in multiple cells lines including HeLa cells and brain cells in a process called 'anastasis' (*Sun et al., 2017*; *Tang et al., 2012*), suggesting that this may be a general mechanism to cope with cellular stress. We attempted to track the fate of individually damaged mitochondria in a localized region after laser treatment (data not shown), but the continuous photoirradiation required to achieve a high spatio-temporal resolution also led to a quick decrease in MitoTracker Red signal and undesirable additional stress to the cell over time,

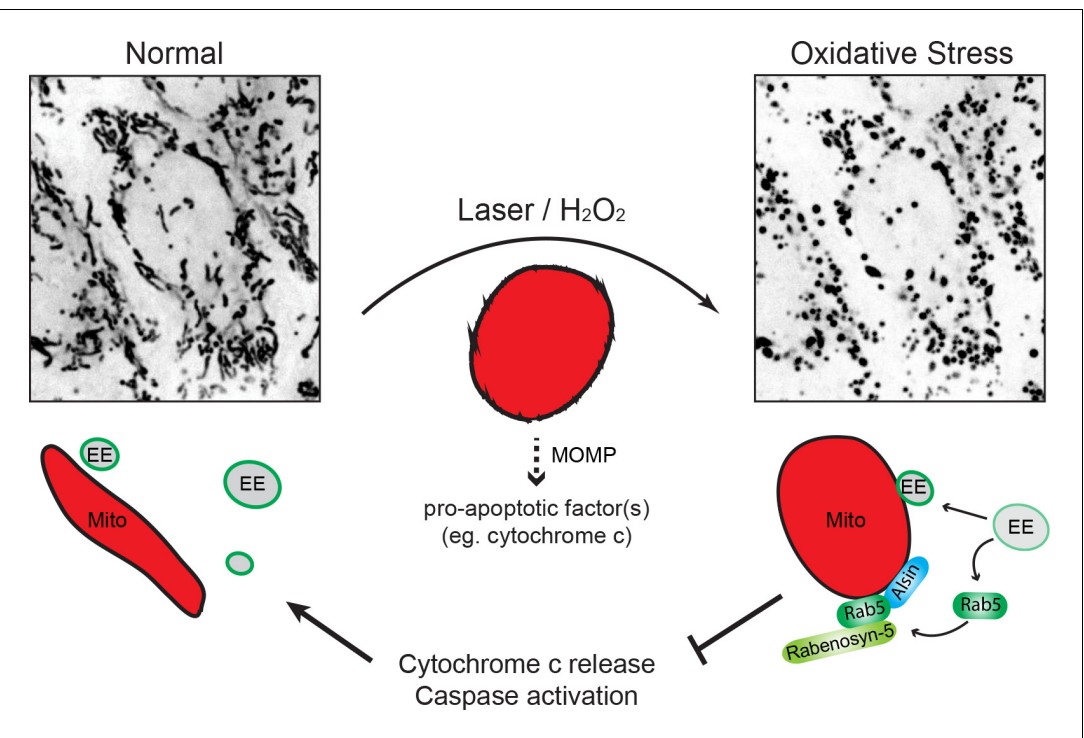

**Figure 12.** Schematic model depicting the role of Rab5-Alsin-mitochondria during oxidative stress. In the normal condition, mitochondria (Mito, red) are elongated and tubular (left). Rab5 (green) are localized on early endosomes (EE) to assemble the Rab5 machinery for endosomal maturation and membrane trafficking. At steady state, some EE make transient contacts with mitochondria. During oxidative stress (e.g. laser- or chemically-induced), mitochondria undergo MOMP and a dramatic morphological transformation into rounded and swollen structures (right). The release of the apoptotic factor, cytochrome c, from mitochondria into the cytosol is associated with a re-localization event of Rab5 from EE to mitochondria via a cytosolic intermediate, accompanied by an increase in EE-mitochondria MCS. The recruitment and activation of Rab5 on mitochondria depend on the Rab5 GEF Alsin (blue), which leads to a selective recruitment of Rabenosyn-5 (light green). This signaling cascade on mitochondria is a reversible process that regulates the apoptotic program (e.g. cytochrome c release and caspase activation) and thus, promotes overall cell survival.
DOI: https://doi.org/10.7554/eLife.32282.050

preventing us from determining its precise outcome. The loss of Mito-Red signal is likely due to MOMP, as evident by the release of cytochrome c, and not a result of mitochondrial clearance because the outer mitochondrial membrane can be stained by TOM20 and visualized by the presence of Rab5 ring-like formation.

Which molecular mechanism is responsible for the dissociation of Rab5 from EE and its recruitment to mitochondria? On EE, the levels of Rab5 depend on the equilibrium between the cytosolic pool of Rab5 complexed to Rab GDI and the membrane-associated pool sustained by the Rabex-5/Rabaptin-5 complex and a plethora of Rab5 effectors (*Del Conte-Zerial et al., 2008*; *Lippé et al., 2001*; *Zerial and McBride, 2001*; *Zhang et al., 2014*). Prior studies have shown that such equilibrium, and consequently endocytic trafficking, is adaptive to stress and apoptotic signal. For example, Rabaptin-5 is selectively cleaved by caspase-3 during apoptosis, thus affecting its interaction with Rab5 and reducing the overall endocytic capacity (*Cosulich et al., 1997*; *Swanton et al., 1999*). The activation of p38 MAPK by $H_2O_2$ also stimulates the formation of the GDI:Rab5 complex, thus favoring the extraction of Rab5 from the early endosomal membrane (*Cavalli et al., 2001*). In addition, p38 MAPK modulates the endosomal function via phosphorylation and membrane association of Rab5 effectors (*Macé et al., 2005*). Such a mechanism may account for the mobilization of Rab5 from the endosomal membrane, which is suggested by increased in cytosolic Rab5 and decreased colocalization of Rab5 with the endosomal EEA1 (*Figure 6A–C*). These phenotypes correlated to a defect in endosomal sorting (*Figure 1E*) and a block in transferrin uptake (*Figure 6D,E*), which may

be yet another protective mechanism in order to avoid iron overload and toxicity associated with neurodegeneration (*Núñez et al., 2012*). Interestingly, hippocampal HT-22 neurons exposed to excess iron exhibit mitochondrial fragmentation and a decrease in cell viability (*Park et al., 2015*). The metabolic effect of $H_2O_2$ on OCR is intriguing. Mitochondria require oxygen to produce ATP to drive energy-consuming reactions (*Bratic and Trifunovic, 2010*). The observed decrease in OCR (*Figure 5E*) may serve to lower cellular respiration and prevent further ROS production.

If the stress response triggers solubilization of Rab5 from EE, then it must also catalyze the activation of Rab5 on mitochondria. We discovered that this step depends on Alsin. The C-terminal VPS9 domain of Alsin has GEF activity towards Rab5, and plays a role in Rab5 endosomal localization and dynamics (*Otomo et al., 2003*; *Topp et al., 2004*). However, the physiological role of Alsin with respect to Rab5 and endosomal activity has remained somewhat mysterious. At steady state, Alsin is mainly cytosolic with a fraction localizing to vesicular-like structures (*Figure 10—figure supplement 1C*) (*Millecamps et al., 2005*). In stress-induced conditions, however, Alsin re-localizes to mitochondria. The N-terminal RLD of Alsin has been shown to exhibit an autoinhibitory effect on its VPS9 domain (*Otomo et al., 2003*). We posit that mitochondrial-induced stress triggers structural changes in the protein, releasing the autoinhibitory effect of RLD, thereby exposing the VSP9 domain for Rab5 activation and recruitment. Alsin was required for Rab5 translocation to mitochondria as this was severely diminished in Alsin$^{-/-}$ iPS-sMN cells. The presence of (low levels) Rabex-5 on mitochondria (*Figure 10A,C*) detected in HeLa cells suggests that other GEFs may contribute to some Rab5 activation, depending on the cell types, but cannot fully compensate for the loss of Alsin function. A homologous gene, ALS2CL, containing only the carboxyl-terminal half of ALS2, may also play a role by specifically binding to Rab5 and forming a homodimer with the full-length Alsin to membranous compartments (*Hadano et al., 2004*; *Suzuki-Utsunomiya et al., 2007*).

What is the function of the assembly of Rab5 and Rab5 machinery (Alsin and Rabenosyn-5) on mitochondria besides its protective role? The stress response triggers remodeling of mitochondria to confer molecular features characteristic of the endocytic system. The Rab5 machinery may be used to bring mitochondria in close proximity to EE and form MCS (*Figure 1*). These MCS may mediate the transfer of lipids and metabolites (*Helle et al., 2013*), or involve in 'patching' up mitochondrial wounds by recruiting the ESCRT machinery for closure, or other endomembranes for fusion, both of which are observed in the PM-repair response (*Jimenez et al., 2014*; *Reddy et al., 2001*). Worth noting from our EM study, we observed ER-like membranous structure in contact with stressed mitochondria (*Figure 3C*) and considering the role of ER-mitochondria contacts in $Ca^{2+}$-regulated apoptosis (*Pinton et al., 2008*), one may postulate that an orchestrated three-way organelle crosstalk exists.

Considering that Rab5 is necessary for the biogenesis of the endolysosomal system (*Zeigerer et al., 2012*), the Rab5 translocation may be a priming step of a stress response pathway that subjects mitochondria to interact with the entire endolysosomal system, in order to modulate the mitochondrial apoptotic potential. One quality control mechanism is the formation of mitochondrial-derived vesicles (MDVs), which are involved in the transport of oxidized or damaged cargo to LE and lysosomes for degradation (*Soubannier et al., 2012*). This process depends on PINK1/Parkin (*McLelland et al., 2014*), but can also occur independently (*Matheoud et al., 2016*). Rab5 may play a role in MDV formation, although we could not detect vesicle budding events in our experimental conditions. Once recruited onto mitochondria, Rab5 activity may not be limited to the recruitment of its effectors, but initiate a more extensive endosomal Rab cascade via the Rab coupling/conversion mechanism. On EE, Rab5 interacts with divalent effectors, coupling its activity to other Rab proteins (e.g. Rab4, Rab11) that are required for receptor recycling (*de Renzis et al., 2002*; *Vitale et al., 1998*). Rab5 also initiates the activation of Rab7, resulting in the conversion of EE into LE (*Rink et al., 2005*). The Rab coupling/conversion may also be initiated on the mitochondria. Therefore, it is possible that the mitochondria-endosome MCS may evolve over time leading to a Rab7-dependent mitophagic pathway (*Jimenez-Orgaz et al., 2018*), the engulfment of mitochondria by the EE (*Hammerling et al., 2017*), or conventional autophagic processes (*Ao et al., 2014*; *Stolz et al., 2014*). Future work exploring the dynamics of other endosomal Rab GTPases in relation to Rab5 will be necessary to elucidate the precise role of the endosomal system on mitochondria.

The physiological role of Alsin, although elusive, has been linked to both endosomes and mitochondria. Cultured hippocampal neurons from Alsin knockout mice display an accumulation of enlarged Rab5 endosomes and a reduced endosomal motility (*Lai et al., 2009*). Mutational and

linkage analysis of Alsin from human patients show that the VPS9 domain is critical for Alsin function (*Daud et al., 2016*; *Verschuuren-Bemelmans et al., 2008*). A recent EM study on the corticospinal motor neurons (CSMN) from Alsin KO mice reveals a selective morphological defect in mitochondria with enlarged core and broken cristae (*Gautam et al., 2016*). Interestingly, WT vs Alsin KO CSMN show no change in Parkin expression, suggesting that mitophagy does not play a major role (*Gautam et al., 2016*). We postulate that the pathological condition of mitochondrial defects in Alsin KO cells is related to a deficiency in Rab5 recruitment to mitochondria, thereby leading to a decline in protection from ROS and oxidative stress associated with aging. In ALS patients, motor neurons likely accumulate more damaged mitochondria as age progresses, which eventually become an over-burden for cells.

The primary cause for ALS is still unclear, but oxidative stress is considered to be a major contributor. Mutations in the antioxidant enzyme, superoxide dismutase 1 (SOD1), are associated with motor neuron degeneration. In mouse models, an accumulation of the SOD1 mutant proteins results in mitochondrial swelling and increased oxidative damage (*Jaarsma et al., 2001*). Interestingly, loss of Alsin in the mutant SOD1 transgenic mice exacerbates and accelerates disease progression (*Hadano et al., 2010*). These studies, along with our findings, corroborate the protective role of Alsin during oxidative stress. The mechanistic link between Rab5 and Alsin may present a general or related mechanism in other neurodegenerative diseases. In Parkinson disease, the most common mutation in the multidomain Leucine-rich repeat kinase 2 (LRRK2) protein leads to a hyper-activation of the kinase domain, resulting in hyper-phosphorylation of a number of Rab GTPase substrates including Rab5 (*Steger et al., 2017*; *Steger et al., 2016*). This may present yet another mechanism of regulating Rab5 localization and function on mitochondria. Future work using different neurodegenerative disease models in differentiated human neurons will provide deeper insights into the disease etiology.

# Materials and methods

## Cell lines, cell culture, and growth conditions

The following cell lines have been validated and tested negative for mycoplasma contamination: HeLa (Kyoto) cell line, BAC HeLa GFP expressing cell lines, and human KOLF_C1 iPSC (kindly provided by Bill Skarnes, Sanger Institute). HeLa cells were cultured in high-glucose DMEM (Gibco) with 10% fetal bovine serum, 100 U/ml penicillin, 100 µg/ml streptomycin, and 2 mM glutamine (all reagents from Sigma-Aldrich) with 5% $CO_2$ at 37°C. All plasmids were transfected using Effectene transfection reagent (Qiagen, Germany) according to the manufacturer's protocol. All bacterial artificial chromosome (BAC) transgene HeLa cell lines expressing different markers were obtained from the BAC recombineering facility at MPI-CBG (Dresden, Germany) and generated using the method previously described (*Poser et al., 2008*).

## Plasmids and chemical reagents

Construction of the pEGFP-C3-2xFYVE was made using mouse Hrs FYVE domain containing a linker (QGQGS) (*Raiborg et al., 2001*). Human Rab5c cDNA was subcloned into the pEGFP-C3 plasmid (Addgene). Human Alsin cDNA subloned into the pEF1/Myc-His (Invitrogen) plasmid was a kind gift from Dr. Ikuo Nishimoto (*Kanekura et al., 2004*). Alexa-conjugated transferrin (Invitrogen; T13342) and EGF (Invitrogen; E13345) were used at 25 µg/ml and 2 µg/ml, respectively. Carbonyl cyanide 3-chlorophenylhydrazone (CCCP) was purchased from Sigma Aldrich (C2759). Stock solution was prepared to a final concentration of 10 mM in DMSO. 100 mM Hydrogen peroxide ($H_2O_2$) (Merck Millipore; 7722-84-1) stock solution was prepared in PBS.

## Live-cell imaging

Cells were seeded either in a 35-mm glass-bottom dish or ibidi µ-Dish 35 mm, high Grid-500. Before imaging, medium was replaced with HEPES-buffered DMEM without phenol red (Gibco). Time-lapse imaging was performed using the Nikon TiE inverted stand microscope equipped with spinning disc scan head (CSU-X1; Yokogawa), fast piezo objective z-positioner [Physik Instrumente], and back-illuminated EMCCD camera (iXon EM +DU-897 BV; Andor). Imaging was done with an Olympus UPlan-SApo 100 × 1.4 Oil and Nikon Apo 100 × 1.49 Oil DIC 0.13–0.20 objectives (illumination by lasers:

DPSS-488nm, DPSS-561nm, DPSS-640nm). Individual planes were recorded at ~11 frames/s with Z-stacks of three planes (step 0.3 μm).

## Photosensitization of mitochondria

Cells were incubated with MitoTracker Red CMXRos (ThermoFisher; M7512) at a final concentration of 100 nM at 37°C for 30 min, 5% $CO_2$ incubator, and followed by 2X PBS wash before irradiating with 561 nm laser on the spinning disc Andor-Olympus-IX71 at a low-power dosage of ~5 J/cm$^2$ for 60 s.

## Correlative light electron microscopy

Cells were grown on a gridded dish (ibidi μ-Dish 35 mm, high Grid-500). Cells in different locations were laser-treated with 561 nm laser for 30 s. Cells were fixed in 2.5% glutaraldehyde/PBS for 30 min at room temperature. Post-fixation and embedding were performed using 1% osmium tetroxide/1.5% potassium ferrocyanide and Epon Lx112, respectively. Sectioning of 150 nm thick UA sections was performed on a Leica Ultracut UCT (Leica Microsystem, Wetzlar, Germany) with a diamond knife. Samples were post-stained with 2% uranyl acetate and lead citrate. 2D images were acquired on a Tecnai T12 (FEI, Hillsboro, OR).

## Immunofluorescence and antibodies

Cells were seeded on a ibidi Grid-500 glass bottom. After laser or $H_2O_2$ treatment, cells were fixed in 4% paraformaldehyde/PBS for 15 min at room temperature. Cells were washed twice with PBS and permeabilized in PBS containing 0.1% saponin, and 1% BSA for 30 min at room temperature. Cells were immunostained with the corresponding primary antibodies: anti-rabbit Rabenosyn-5/ZFYVE20 (Sigma Aldrich: HPA044878), anti-mouse EEA1 (BD Biosciences: 610457), anti-rabbit TOM20 (Santa Cruz Biotechnology: sc-11415), anti-rabbit APPL1 (Abcam: ab59592), anti-mouse APPL2 (home-made), anti-mouse Rab5 (BD Biosciences: 610724), anti-mouse cytochrome c (Abcam: ab6311), and anti-rabbit Alsin (Novus Biological: NBP2-14284) antibodies. Alexa fluor-conjugated (ThermoFisher) were used as secondary antibodies. Samples were mounted with Mowiol (Sigma-Aldrich) on glass slides and examined using the Zeiss LSM 880 inverted single photon point scanning confocal system with Quasar detector (32 spectral detection channels in the GaAsP detector plus 2PMTs) and transmitted light detector. Acquired images were processed and saved using the Zeiss ZEN software. For immunofluorescence on iPSCs, smNPCs, and sMNs, cells were fixed with 4% formaldehyde for 15 min, washed three times with wash buffer (0.3% Triton-X in PBS) for 5–10 min, and blocked with blocking buffer (5% goat serum, 2% BSA, and 0.3% Triton-X in PBS) at room temperature for 1 hr. Cells were incubated with primary antibodies in blocking buffer overnight at 4°C. After three washes with PBS for 10 min, cells were incubated with secondary antibodies in wash buffer for 2–3 hr at room temperature followed by three washes in PBS for 10 min. Primary antibodies used include: goat anti-ChAT (1:100) (Millipore, #AB144P), mouse anti-HB9 (1:50) (DSHB, #81.5C10, conc.), rabbit anti-ISL1 (1:100) (Abcam, #ab20670), mouse anti-LIN28 (1:1000) (Cell signaling, #5930S), chicken anti-MAP2 (1:1000) (Novus Biologicals, #NB300-213), mouse anti-Nestin (1:150) (R and D Systems, #MAB1259), rabbit anti-OCT4 (1:500) (Abcam, #ab19857), rabbit anti-PAX6 (1:300) (Covance, #PRB-278P), and rabbit anti-SOX2 (1:500) (Abcam, #ab97959). Ad-Ctrl and Ad-tBid (a kind gift from Dr. Heidi McBride) were used at 1:200 PFU/cell.

## Transferrin uptake

Cells were seeded in a 384-well plate and incubated with either complete medium or in the presence of 250 μM $H_2O_2$ at 37°C for 2 hr. Cells were then pulsed with Alexa-647 Tfn (10 μg/ml) for 5 min, followed by 3x PBS wash, fixed with 3.7% PFA for 15 min, and then stained with DAPI (1:1000) and CellMask Blue (1:2000) (ThermoFisher). Image acquisition was performed via the automated confocal imaging system, CV7000S Yogokawa. Images analysis were performed using MotionTracking software.

## Subcellular fractionation

Cytosolic and mitochondrial fractions were performed using the mitochondria isolation kit, according to the manufacturer's protocol with minor modification (ThermoFisher: cat89874). Cells (~$1\times10^7$)

were resuspended in 400 µl Mitochondrial Isolation Reagent A. Cells were chemically lysed by adding 5 µl of Reagent B. After 5 min incubation on ice, 400 µl of Reagent C was added to each sample and centrifuged at 720 x g for 10 min. The post-nuclear supernatant (PNS) was transferred to a new eppendorf tube and centrifuged at 3000 x g for at 4°C for 15 min. For the total membrane and purified cytosolic fractions, the PNS was clarified at 200,000 g at 4°C for 1 hr. The resulting supernatant was collected and trichloroacetic acid (TCA)/acetone precipitation was performed to obtain the final cytosolic fraction. The remaining pellet was washed by adding 500 µl of Reagent C and centrifuged at 15,000 x g for 5 min. Final samples were resuspended in SDS loading buffer.

## Cytochrome c release assay and western blot

Cells were seeded on a 12-well plate. For hydrogen peroxide treatment, reagent was added directly into the well to achieve the appropriate concentration. Separation of mitochondrial and cytosolic fractions were performed using the mitochondrial isolation kit from ThermoFisher (cat:89874) with an additional step of trichloroactic acid precipitation of the cytosolic fraction. The final pellet was dried in a 95°C heat block for 2–3 min before resuspending it in the SDS loading buffer. Cell lysates were separated by SDS-PAGE, transferred onto the nitrocellulose membrane and blocked in 5% milk in PBS containing 0.1% Tween. Primary and secondary antibodies were diluted in the blocking buffer and incubated at room temperature for 2 hr. The bands were detected using the electrochemiluminescence reagent and exposure onto x-ray films. The following antibodies were used in western blot: anti-mouse cytochrome c (Abcam: ab13575), anti-rabbit gamma tubulin (Sigma-Aldrich: T6557), anti-rabbit Rabenosyn-5/ZFYVE20, anti-rabbit Alsin (Sigma Aldrich: SAB4200137), anti-mouse EEA1 (BD Biosciences: 610457), anti-rabbit APPL1 (Abcam: ab59592), anti-mouse APPL2 (home-made), anti-mouse GAPDH (Sigma Aldrich: G8795), anti-mouse gamma tubulin (Sigma Aldrich: T6557), and anti-rabbit TOM20 (Santa Cruz Biotechnology: sc-11415).

## The caspase-3/7 activation assay

The caspase-3/7 activation in CCCP- vs $H_2O_2$- treated cells were measured using the caspase-3/7 green flow cytometry assay kit (ThermoFisher: C10427). After 2 hr treatment, cells were scraped off per well from a six-well dish, gently resuspended in 1 ml of PBS containing 1 µl of green detection reagent, and incubated at 37°C for 20 min. Samples were analyzed using the 488 nm excitation with standard fluorescence compensation and emission filter (530/30 BP) in the FACS Calibur (Beckton Dickinson). The gating was set based on the background signal in the DMSO control. The total cell count was set to 5000. The caspase-3/7 activation in Alsin and Rab5 over-expressed cells was measured using the caspase-3/7 green detection reagent (ThermoFisher: C10423). Cells were incubated in a complete medium containing 5 µM of green detection reagent at 37°C for 30 min prior to fixation. Cells were imaged using the Zeiss LSM 880 microscope.

## Measurements of mitochondrial oxygen consumption rates

20,000 HeLa cells per well were seeded in a XFe96-well plate (Seahorse Bioscience), and grown to ~80% confluency after overnight incubation. Cells were then equilibrated with a bicarbonate-free DMEM medium containing 4 mM glutamine and 10 mM galactose in a 37°C ambient $CO_2$ incubator for 1 hr, before starting the experiment. CCCP and $H_2O_2$ compounds were prepared fresh and diluted in the assay medium, and were injected from the reagent ports at the indicated time. Oxygen consumption rates (OCR) were measured using a Seahorse Extracellular Flux Analyzer.

## Generation of CRISPR/Cas9 knockout in human induced pluripotent cells

Human KOLF_C1 iPSC were cultured in feeder-free conditions on Matrigel with TeSR-E8 media (StemCell, Germany). For ALS2/Alsin knockout using CRISPR/Cas9 genome editing, 350,000 cells were detached using Accutase, washed once with PBS and electroporated using the Neon Transfection System (Invitrogen, Germany, 10 µl kit, 1000V, 20 ms, three pulses). The genomic sequence of human Alsin was analyzed for CRISPR/Cas9 target sites by Geneious 8.1.6 software (Biomatters), and two pairs of guides flanking a critical exon (exon3) were selected (5'-GCTAAAGTACTGAA TTTTGG-3' and 5'-AATAAAATCAGCAGGTGTGG-3'; 5'-GAATTTCTACAAAGTGCAGG-3' and 5'-TAGCCTGGATGATGGCCGTT-3') and were used together to cause a frame shifting exon deletion.

The in vitro efficiency of these gRNAs was assessed by generating a genomic PCR cleavage template of 3.4 bp (primers used: for-CCTCCCTTCCCAGGATCTGA and rev-TGCTCAACTCGAGTGCCTTT; for-CAGGGTGAGCATCCCACATT and rev-AGGAGTTCCAGTCAACCAGT) and incubating with recombinant Cas9. All gRNAs used in vitro were identical in sequence to the DNA sense strand and not complementary to the mRNA sequence. The RNAs employed in this method were chemically modified and length optimized variants of the native guide RNAs (Alt-R[TM] CRISPR crRNAs and tracrRNA, Integrated DNA Technologies, Coralville, IA). The recombinant Cas9 (provided by Protein Expression Facility at MPI-CBG) protein from *Streptococcus pyogenes* was used. The crRNAs were mixed with trRNA and NLS-Cas9 (1 µg/µl). The guide RNA complex was formed by mixing the crRNAs and tracrRNAs in equal amounts in Buffer R (Invitrogen, Germany) at 100 µM. Five days after electroporation, cells were pooled and seeded for clonal dilution. Single clones were mechanically picked and amplified. Next, genomic DNA was isolated using QuickExtract DNA Extraction Solution (EpiCentre, USA). Homozygous deletions were verified by PCR and sequencing.

## Generation of iPSC-derived smNPC and differentiation of smNPCs to sMNs

All procedures were performed as previously described (*Reinhardt et al., 2013*). Briefly, for smNPC generation, iPSC colonies detached from Matrigel-coated wells (by 1 mg/ml dispase) were resuspended in hESC medium (DMEM/F12, 20% KnockOUT Serum Replacement, 1% Penicillin/Streptomycin/Glutamine, 0.1 mM Non-Essential Amino Acids Solution, 0.05 mM beta-mercaptoethanol, without bFGF supplemented with 10 µM SB431542 (Tocris, #1614), 1 µM dorsomorphin (Tocris, #3093), 3 µM CHIR99021 (Axon Medchem, #Axon-1386) and 0.5 µM purmorphamine (STEMCELL Technologies, #72202), and cultured in non-coated petri dishes. After 2 days, hESC medium was replaced by N2B27 medium (1:1 mixture of DMEM/F12 and Neurobasal medium, 1% Penicillin/Streptomycin/Glutamine, 1:100 B-27 supplement minus vitamin A, 1:200 N-2 supplement) and supplemented with the same small molecules as listed above. After another 2 days, culture medium was replaced by smNPC expansion medium (N2B27 medium supplemented with 150 µM ascorbic acid (Sigma, #A4403), 3 µM CHIR99021 and 0.5 µM purmorphamine). On day 6 of neural induction, embryonic bodies were broken into smaller clumps by titration and plated in six wells of a Matrigel-coated 12-well plate. On day 9, cells were passaged for the first time using Accutase at a 1:3 split ratio and seeded in four wells of a Matrigel-coated six-well plate. Afterwards, cells were passaged once a week and seeded at a density of $1 \times 10^6$ cells per well. To obtain a highly pure smNPC culture, smNPCs were propagated for at least 10 passages in smNPC expansion medium. For differentiation of smNPC to MNs, smNPCs were seeded at a density of $1.5 \times 10^6$ cells per one well of a Matrigel-coated six-well plate and cultured in N2B27 medium supplemented with 1 µM purmorphamine for the first 2 days of differentiation. The cells were then cultured in N2B27 medium supplemented with 1 µM purmorphamine and 1 µM retinoic acid (Sigma, #R2625) until day 9 of differentiation. On day 9, cells were dissociated using Accutase and plated on polyornithine/laminin-coated ibidi µ-slides (at a density of 150000 cells per well) or Nunc four-well plates (at a density of 300,000 cells per well) in maturation medium (N2B27 medium supplemented with 0.5 mM cAMP (Sigma, #D0627), 10 ng/ml BDNF (Peprotech, #450-02-10), and 10 ng/ml GDNF (Peprotech, #450-10-10)). Cells were maintained in maturation medium until analysis on day 28.

### Image and statistical analysis

Image resizing, cropping and brightness were uniformly adjusted in Fiji (http://fiji.sc/). Colocalization analysis was performed using MotionTracking software (*Rink et al., 2005*) (http://motiontracking.mpi-cbg.de/get/) and described previously (*Gilleron et al., 2013*). The y-axis is expressed as the ratio of co-localized objects (e.g. A to B) to total objects found in A. Final images were assembled using Adobe Photoshop and Illustrator. Densitometry quantification were performed in Fiji following the previously described protocol (http://www.yorku.ca/yisheng/Internal/Protocols/ImageJ.pdf). p Values were calculated by two-tailed t-test using GraphPad Prism7.

## Acknowledgements

We thank the MPI-CBG light microscopy facility for access and technical assistance; the TransgeneOmics facility and Hyman lab, especially Mihail Sarov, Aleksandra Syta, Kathleen Rönsch, Marit

Leuschner, and Ina Poser, for the design, generation, and maintenance of the CRISPR/Cas9 KO iPSC lines; Julia Japtok from the lab of Andreas Hermann for sharing protocols for the smNPC generation and MN differentiation; Weihua Leng and Urska Repnik for technical assistance and method discussion for electron microscopy; Ina Nuesslein and Christina Eugster from the MPI-CBG FACS facility for assistance on the flow cytometry; Rico Barsacchi from the Technology Development Studio facility for assistance with the transferrin uptake and Seahorse Analyzer; Yannis Kalaidzidis for assistance with MotionTracking software; and Heidi McBride for reagents, discussion and feedback.

## Additional information

### Competing interests

Anthony A Hyman: Reviewing editor, *eLife*. The other authors declare that no competing interests exist.

### Funding

| Funder | Grant reference number | Author |
| --- | --- | --- |
| Human Frontier Science Program | LT000225/2016-L | FoSheng Hsu |
| EMBO | ALTF 923-2015 | FoSheng Hsu |
| Max-Planck-Gesellschaft | Open-access funding | Marino Zerial |

The funders had no role in study design, data collection and interpretation, or the decision to submit the work for publication.

### Author contributions

FoSheng Hsu, Conceptualization, Resources, Data curation, Software, Formal analysis, Supervision, Funding acquisition, Validation, Investigation, Visualization, Methodology, Writing—original draft, Project administration, Writing—review and editing; Stephanie Spannl, Data curation, Formal analysis, Methodology, Writing—review and editing; Charles Ferguson, Data curation, Investigation, Methodology; Anthony A Hyman, Supervision, Writing—review and editing; Robert G Parton, Data curation, Investigation, Methodology, Writing—review and editing; Marino Zerial, Conceptualization, Supervision, Funding acquisition, Investigation, Writing—original draft, Writing—review and editing

### Author ORCIDs

FoSheng Hsu http://orcid.org/0000-0001-8486-7311
Robert G Parton http://orcid.org/0000-0002-7494-5248
Marino Zerial http://orcid.org/0000-0002-7490-4235

### Decision letter and Author response

Decision letter https://doi.org/10.7554/eLife.32282.053
Author response https://doi.org/10.7554/eLife.32282.054

## Additional files

### Supplementary files

• Transparent reporting form
DOI: https://doi.org/10.7554/eLife.32282.051

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
