## [Decision Letter]

Thank you for submitting your article "Rab5 and Alsin regulate stress-activated cytoprotective signaling on mitochondria" for consideration by *eLife*. Your article has been reviewed by three peer reviewers, and the evaluation has been overseen by a Reviewing Editor and Ivan Dikic as the Senior Editor. One of the reviewers, Nicholas T Ktistakis, has agreed to reveal his identity.

The reviewers have discussed the reviews with one another and the Reviewing Editor has drafted this decision to help you prepare a revised submission.

The manuscript of Hsu et al. focuses on the role of Rab5 and Alsin in mitochondrial protection during oxidative stress. The authors find that Rab5 translocates toward the mitochondrial surface under laser-induced oxidative stress, which seems to occur at contact sites between endosomes and mitochondria. Under these conditions, mitochondria do not become substrates for mitophagy, but are rather protected from degradation. In agreement with this statement, cells lacking Alsin are more susceptible to cytochrome c release and thus cannot presumably take advantage of the Rab5-mediated protection.

The reviewers unanimously found this work to be very interesting, pointing out that the translocation of Alsin and Rab5 onto mitochondria upon oxidative stress was completely unexpected, but that the story was nevertheless very convincing. However, they also highlighted two areas where they felt more work was needed: further exploration of the mechanism of Alsin/Rab5 translocation, and more quantitative data analysis.

*Quantification:*

1) The authors highlight the transient contacts between mitochondria and endosomes (Figure 1 and videos), but this is not quantified in any meaningful way. In any case, the colocalization itself does not demonstrate a functional coupling, as the authors themselves state. Do the authors see differential interactions between EGF positive endosomes compared to transferrin? The videos are done primarily with transferrin, but what about the EGF endosome? A more careful comparison of these two types of endosomes could help back up their statement that these observed interactions may be specific. Similarly, it's hard to understand whether the apparent contact sites seen during stress are really significantly increased relative to steady state – or what function that may have. Even in the Discussion the authors state that the contact sites "appear to increase upon stress", but this seems to be a "feeling" of the authors without concrete quantification.

2) Not all experiments are quantified, and we are sometimes looking at perhaps one cell over a time course (as in Figure 2). The major observations have been quantified, as in Figures 1 and 4, but this is a single time point and treatment. An analysis of recruitment over time is needed so we can better understand the kinetics – which is mechanistically very important. The CLEM shows one cell, and there is no negative control. While the technique is appreciated, more informative, quantitative EM could be done with straight immunoEM for the GFP-Rab5 showing recruitment to mitochondria. Again, with a more quantitative EM approach they could also quantify ER contacts, which mediate calcium flux from the ER into mitochondria during almost all death paradigms, again correlating this to Rab5 recruitment.

*Mechanism:*

1) The reviewers were not convinced that the formation of contact sites between endosomes and mitochondria was all that relevant to the translocation of Alsin and Rab5. Finding out how Rab5 ends up on the mitochondria (vesicular delivery or translocation from the cytosol) is important.

Deciding on whether this is a Rab5 moonlighting function as opposed to an integrated response of the endocytic system to oxidative stress and accordingly modifying the tone of the text is also critical. A number of experiments were suggested, but upon discussion the reviewers agreed that some of them were beyond the scope of this paper.

2) There were also several more technical comments from reviewers related to the fact that Rab5 is recruited during peroxide and irradiation damage, but not with CCCP – and that they do not observe mitophagy. The focus of a revision should be on sorting the links to the apoptotic program.

---

## [Author Response]

[…] The reviewers unanimously found this work to be very interesting, pointing out that the translocation of Alsin and Rab5 onto mitochondria upon oxidative stress was completely unexpected, but that the story was nevertheless very convincing. However, they also highlighted two areas where they felt more work was needed: further exploration of the mechanism of Alsin/Rab5 translocation, and more quantitative data analysis.

Quantification:

1) The authors highlight the transient contacts between mitochondria and endosomes (Figure 1 and videos), but this is not quantified in any meaningful way. In any case, the colocalization itself does not demonstrate a functional coupling, as the authors themselves state. Do the authors see differential interactions between EGF positive endosomes compared to transferrin? The videos are done primarily with transferrin, but what about the EGF endosome? A more careful comparison of these two types of endosomes could help back up their statement that these observed interactions may be specific. Similarly, it's hard to understand whether the apparent contact sites seen during stress are really significantly increased relative to steady state – or what function that may have. Even in the Discussion the authors state that the contact sites "appear to increase upon stress", but this seems to be a "feeling" of the authors without concrete quantification.

We internalized fluorescent Tfn and EGF for different periods of time and quantified the colocalization between Tfn+ and EGF+ endosomes and mitochondria during normal and stressed conditions. We chose to assess the interaction with fixed cells, rather than by live-cell imaging to obtain a statistically significant dataset on a population of cells. Our results indicate a clear preferential interaction of mitochondria with Tfn+ endosomes. This result not only helped us to validate our initial observation of increased endosomal-mitochondrial MCS upon stress but also pinpointed the preferential interaction with early endosomes/recycling endosomes (new Figure 1D, E). The functional significance of these MCS are further discussed in detail under the revised “Discussion” section. Furthermore, our quantifications do confirm the increase in Rab5-positive vesicles docked onto following Rab5 translocation (Video 6 and new Video 7), suggesting that Rab5 translocation and endosomes-mitochondria docking are sequentially linked.

2) Not all experiments are quantified, and we are sometimes looking at perhaps one cell over a time course (as in Figure 2). The major observations have been quantified, as in Figures 1 and 4, but this is a single time point and treatment. An analysis of recruitment over time is needed so we can better understand the kinetics – which is mechanistically very important.

We have included new data showing the kinetics of Rab5 recruitment to mitochondria by time-lapse video microscopy (new Video 5). Quantification of the signal intensity and corresponding fluorescent images are presented in the new Figure 2C, D, E. These and other data (see below) also support the interpretation that Rab5 is not delivered by endosome fusion but recruited from the cytosol.

The CLEM shows one cell, and there is no negative control. While the technique is appreciated, more informative, quantitative EM could be done with straight immunoEM for the GFP-Rab5 showing recruitment to mitochondria. Again, with a more quantitative EM approach they could also quantify ER contacts, which mediate calcium flux from the ER into mitochondria during almost all death paradigms, again correlating this to Rab5 recruitment.

The purpose of the CLEM was not to repeat the quantitative LM analysis at the EM level but to examine the ultrastructure of a structure *representative* of those characterized in our extensive real-time light microscopic study. Using our CLEM/serial sectioning approach we could describe the ultrastructure of the representative Rab5-positive structure and define the proximity to, and morphology of, the adjacent mitochondrion. We believe the CLEM is therefore an important complement to the fixed and real-time quantitative light microscopy.

The complex serial section CLEM technique, which was required to identify the mitochondrial-associated endosomes that were visualized by LM, is not a feasible approach for the large analyses required for quantification purposes. To address the reviewer’s suggestion for quantitative EM, we used BAC GFP-Rab5 cells either fixed chemically with paraformaldehyde to make Tokuyasu thawed cryosections, or cryo-fixed using high-pressure freezing, followed by embedding in lowicryl resin HM20. Thin Tokuyasu or HM20 resin sections were immuno-gold labeled using three different GFP antibodies, but all turned out to produce very weak labeling. Since in the BAC cell line GFP-Rab5 is expressed only at a fraction of endogenous Rab5 levels (because the BAC technology intentionally avoids overexpression) and immunogold labeling efficiency is generally low, this result is not surprising. Additionally, we labeled sections with anti-Rab5, which also produced weak labeling, and is consistent with our previous difficulties of immunogold labeling on endogenous Rab5 proteins (Chavrier et al., Cell, 1990). Therefore, despite some observable labeling on mitochondria in stress condition, we conclude that the labeling is too weak to perform reliable quantification.

While the reviewers’ comments are insightful on pointing out the potential role of ER-mitochondrial contact sites, we believe further and extensive analysis exceeding the scope of our current work would be needed to fully address this question. Nevertheless, we do consider it an important point and have now addressed it in the Results (subsection “Membrane contacts between Rab5-positive mitochondria and endosomes”) and Discussion (fifth paragraph).

Mechanism:

1) The reviewers were not convinced that the formation of contact sites between endosomes and mitochondria was all that relevant to the translocation of Alsin and Rab5. Finding out how Rab5 ends up on the mitochondria (vesicular delivery or translocation from the cytosol) is important.

In addition to the time-lapsed analysis (Figure 2C-E and Video 5), we have conducted subcellular fractionation experiments to separate the total membrane from the cytosol. We detected an increase of Rab5 in the cytosolic fraction (Figure 6A, B). We also detected a decrease in the levels of endosomal Rab5. Altogether, the data point to a translocation mechanism from early endosomes to mitochondria via a cytosolic intermediate, instead of vesicular delivery. We have revised the text to describe this mechanistic finding (Results subsection “Hydrogen peroxide-induced stress triggers Rab5 translocation from EE to mitochondria, increases early endosomal-mitochondrial contacts, and interferes with transferrin uptake” and Discussion).

Deciding on whether this is a Rab5 moonlighting function as opposed to an integrated response of the endocytic system to oxidative stress and accordingly modifying the tone of the text is also critical. A number of experiments were suggested, but upon discussion the reviewers agreed that some of them were beyond the scope of this paper.

We appreciate the reviewers’ suggestions and consideration, and have addressed many of them experimentally. With the incorporation of new data, we have edited the text throughout the revised manuscript.

2) There were also several more technical comments from reviewers related to the fact that Rab5 is recruited during peroxide and irradiation damage, but not with CCCP – and that they do not observe mitophagy. The focus of a revision should be on sorting the links to the apoptotic program.

To address this mechanistic point, we have performed further analysis on the differential Rab5 recruitment between H_2_O_2_ and CCCP in cells. It is clear from the literature that both compounds exhibit specific effects on mitochondria. However, the effect on mitochondrial respiration is less defined. Given that our mechanism appears to be cytoprotective, we hypothesize that it may do so by slowing down metabolism as a means to conserve energy. Therefore, we set out to measure the oxygen consumption rate (OCR) by mitochondria in H_2_O_2_ vs CCCP conditions, which has led to an interesting and unexpected result (Figure 5E).

In short, these two compounds manifest opposite effects: Whereas CCCP increases OCR, H_2_O_2_ does the converse but recovers over time. To our knowledge, this has not been simultaneously examined in such an experimental set-up. Importantly, we were also able to show that the H_2_O_2_-induced phenotype on Rab5 and mitochondria is reversible (new Figure 8).

In an attempt to examine the link to the apoptotic program, we included the new result of Bax staining in Figure 4, which shows some degree of enrichment in the stressed-induced condition. Additionally, we showed that over-expression of tBid can also induce Rab5 enrichment to mitochondria (Figure 4—figure supplement 4).

Mechanistically, our new findings have also led us to discuss our data with respect to a recently described process called “*anastasis*”, in which cells recover from an apoptotic state normally thought to be irreversible. Our data appear to be in line with this phenomenon (Discussion, second paragraph).